A novel brain tumor magnetic resonance imaging dataset (Gazi Brains 2020): initial benchmark results and comprehensive analysis

Sagiroglu Seref 1
Terzi Ramazan 2 3
Celtikci Emrah 4
http://orcid.org/0000-0002-6222-382X Börcek Alp Özgün 4
http://orcid.org/0000-0002-3298-3334 Atay Yilmaz 1
Arslan Bilgehan 1 bilgehanarslan88@gmail.com
http://orcid.org/0000-0002-5141-8154 Sahin Mustafa Caglar 5
Nernekli Kerem 6
Demirezen Umut 1
http://orcid.org/0000-0002-2829-3672 Ozdemir Okan Bilge 7
Özdem Karaca Kevser 1
Azgınoğlu Nuh 8
1 Computer Engineering, Gazi University , Ankara , Turkey
2 Computer Engineering, Amasya University , Amasya , Turkey
3 Presidency of The Republic of Türkiye, Digital Transformation Office , Ankara , Turkey
4 Medicine, Gazi University , Ankara , Turkey
5 Neurosurgery, Kulu State Hospital , Konya , Turkey
6 Internal Medicine, Icahn School of Medicine , New York City , United States
7 Computer Engineering, Artvin Coruh University , Artvin , Turkey
8 Computer Engineering, Kayseri University , Kayseri , Turkey
Coelho Paulo Jorge
Electronic publication date: 2025 Jun 10
Publication date: 2025
Volume: 11
Electronic Location ID: e2920
Received 2024 Dec 13; Accepted 2025 May 5
Copyright: © 2025 Sagiroglu et al.
Copyright year: 2025
Copyright holder: Sagiroglu et al.
License: This is an open access article distributed under the terms of the Creative Commons Attribution License, which permits unrestricted use, distribution, reproduction and adaptation in any medium and for any purpose provided that it is properly attributed. For attribution, the original author(s), title, publication source (PeerJ Computer Science) and either DOI or URL of the article must be cited.
License URL: https://creativecommons.org/licenses/by/4.0/

Keywords: Anomaly detection, Benchmark, Classification, Deep learning, Gazi Brains 2020, MRI, Prediction, Segmentation, Turkish brain project, Modelling

Funding: Turkish Brain 2021-001 Digital Transformation Office of the Presidency of the Republic of Turkiye This article is a part of the Turkish Brain Project (No: 2021-001) supported by the Digital Transformation Office of the Presidency of the Republic of Turkiye. The funders had no role in study design, data collection and analysis, decision to publish, or preparation of the manuscript.

==============================
This article presents a new benchmark MRI dataset called the Gazi Brains Dataset 2020, containing MRI images of 100 patients, and introduces initial experimental results performed on this dataset in comparison with available brain MRI datasets. Furthermore, the dataset is analyzed using eight different deep learning models for high-grade glioma tumor prediction, classification, and detection tasks. Additionally, this study demonstrates the results of an explainable Artificial Intelligence (XAI) approach applied to the trained models. To demonstrate the utility of the proposed dataset, different deep learning models were applied to the problem, and these models were tested on various data and models applied for various tasks such as region of interest extraction, whole tumor segmentation, prediction, detection, and classification with accuracy, precision, recall, and F1-score. The experimental results indicate that the dataset is highly effective for multiple purposes, and the models reached significant results with successful F1-scores ranging between 93.2% and 96.4%. ROI and whole tumor segmentations were successfully performed and compared with seven algorithms with accuracies of 87.61% and 97.18%. The Grad-CAM model also demonstrated satisfactory accuracy across the tests that were conducted. Moreover, this study explores the application of XAI to the trained models, providing interpretability and insights into the decision-making processes. The findings signify that this dataset holds significant potential for various future research directions, including age estimation, gender detection, causal inference with XAI, and disease-related survival analysis.

Introduction

The identification, classification, diagnosis, and treatment of diseases, along with the systematic organization and refinement of methodologies used in these processes, have required considerable effort over time. The complexity of the human body, the variety of diseases, and the continuous evolution of medical knowledge have rendered these tasks challenging and essential. Over the centuries, advancements in medical science have led to more accurate diagnostic tools, innovative treatment options, and more comprehensive classification systems. However, despite these significant advancements, numerous challenges persist, including the limitations of existing technologies, variability in patient responses to treatment, and the emergence of novel diseases. Nevertheless, ongoing efforts in medical research, the development of more advanced diagnostic technologies, and the refinement of treatment strategies continue to drive improvements in disease management. These continuous advancements are crucial in improving the overall effectiveness, efficiency, and accessibility of healthcare on a global scale (Rahe & Arthur, 1978). As a result, various medical specializations have emerged in this field, and disease analysis processes have become more systematic with the development of tools and approaches suitable for different purposes. Recently, sophisticated imaging and analysis tools have been developed, and effective methods have been proposed to identify lung, liver, heart, and breast diseases, traumatic brain damage, eye and inner ear anomalies, and central nervous system-related problems and to analyze medical images containing tumors. Brain tumors, which represent abnormalities in the brain, are a particularly dangerous situation that can directly affect human health. Brain tumors consist of tissues composed of abnormal cells in the brain, which may cause cancer. Consequently, the early detection and accurate grading of brain tumors are paramount in clinical practice. Several high-cost imaging modalities, including computed tomography (CT), single-photon emission computed tomography (SPECT), positron emission tomography (PET), magnetic resonance spectroscopy (MRS), and magnetic resonance imaging (MRI), are widely employed to gather crucial information regarding tumors, such as their type, shape, size, location, and other essential features necessary for accurate diagnosis and treatment planning in oncology (Díaz-Pernas et al., 2021). Among these modalities, MRI is the most commonly utilized and widely recognized imaging technique in the medical field (Liu et al., 2020; Dhole & Dixit, 2022). MRI scans provide valuable insights for diagnosing a broad spectrum of neurological disorders, including brain tumors, multiple sclerosis (MS), Alzheimer’s disease, Parkinson’s disease, Wilson’s disease, epilepsy, dementia, and autism (Dhole & Dixit, 2022; Ranjbarzadeh et al., 2023; Abd-Ellah et al., 2019; Ozcelik, Altan & Kaya, 2024).

Nowadays, Artificial Intelligence (AI)-based systems have emerged as a prominent area of research, playing a crucial role in various domains, including healthcare, security, and industry. AI is regarded as a transformative technology capable of enhancing problem-solving processes and optimizing task execution across diverse real-world applications. Image generation for the MRI method requires the use of specific imaging sequences, such as T1-weighted, T2-weighted, and FLAIR, which provide different information for experts and AI-based decision support systems to diagnose and identify brain diseases, tumors, or other anomalies (Díaz-Pernas et al., 2021). The noticeable increase in data size and variety is helping scientists to develop better and more accurate models, solve problems from a broader perspective, build new technologies, and make systems and machines more intelligent with the help of big data. Furthermore, the promotion of open data initiatives, collaborative development platforms, and publicly accessible health datasets has played a pivotal role in advancing AI-driven medical research. Utilizing standardized and widely recognized datasets allows for the objective evaluation and comparison of AI models, architectures, and algorithms, ensuring reliability and reproducibility in medical imaging studies.

In recent years, significant attention has been directed toward developing and disseminating datasets to enhance AI algorithms’ ability to model, learn, predict, classify, and ultimately improve their accuracy beyond human-level performance. By using well-established benchmark datasets, researchers have designed and applied novel, high-performance AI algorithms, architectures, and models in various fields, including neurology. The availability of sufficient and well-structured data is a fundamental factor in increasing the success of AI-driven methods and models. The success rate of models has increased, and the potential has been exposed to the presence of structured, meaningful, and high-quality data. Consequently, one of the most critical challenges in this research domain is ensuring access to large-scale, standardized, and well-organized datasets. When these essential conditions are met, the high potential of the models will be revealed, and more successful approaches will be paved.

According to a report published by the Turkish Radiology Association, Türkiye has the highest number of MRI studies per capita among the Organization for Economic Cooperation and Development (OECD) countries (144 per 1,000 persons) (Turkish Radiology Association, 2020; Atici et al., 2020). However, there are only five radiologists per 100,000 people, and the average time allocated for evaluating and reporting radiological studies is approximately 5 min. To improve models, get more knowledge, and do tasks more precisely, effectively, and efficiently, it is vital to organize, standardize, normalize, and benchmark data in an area where the volume and density of data are so high. The dataset introduced in this study was created to address these challenges and contribute to advancements in AI-driven medical imaging analysis.

Brain tumor MRI datasets are essential for evaluating and comparing studies in areas such as brain tumor detection, segmentation, and classification, as well as tumor and treatment progression, brain modeling, MRI preprocessing, age prediction, gender detection, survival analysis, and other topics. Most recent machine learning and deep learning approaches have been developed using MRI images obtained from different MRI scanners produced by different companies. The quality of the datasets is considered a touchstone in the evaluation of the developed methods, and open-access datasets are often preferred, as they do not require additional ethical or clinical approvals/permissions due to having been anonymized or ethically approved in the early stages (Liu et al., 2020). Although several brain tumor MRI datasets exist in the literature, a standard has not generally been established, released, and followed during the generation of these datasets (Tiwari, Srivastava & Pant, 2020). In order to address these problems and support researchers, the Turkish Brain Project was initiated by researchers from the Computer Engineering Department at Gazi University, the Department of Neurosurgery at the Faculty of Medicine, and the Presidency of The Republic of Türkiye Digital Transformation Office (CD-DDO) in 2019. The aims of the project are: (1) designing new AI-based systems to support or assist medical professionals, such as an alert system for real-time diagnosis; (2) developing novel deep learning methods and models for neuroscience and supporting researchers to conduct better and more robust studies, while reducing the cost and time of reporting evaluation processes in MRI images; (3) releasing and deploying highly accurate AI-based commercial tools to support medical experts in clinical practice; (4) innovating key AI technologies, such as Explainable AI (XAI) and next-generation intelligent systems, to further enhance medical diagnosis.

One of the important contributions of the presented dataset is that it has comprehensive labels and standardized data acquired from different MR devices compared to any other open dataset in the literature, and the labeling process is carried out by at least two experts and double-checked. Furthermore, initial tests have demonstrated that the dataset contains sufficient examples for benchmarking purposes. This study also has applied XAI technology, which has been covered in a few research studies but has not yet been applied to brain MRI datasets (Saleem, Shahid & Raza, 2021; Zeineldin et al., 2022).

The article is organized as follows: “Brain Tumor MRI Datasets Available” provides a comprehensive review of the existing brain MRI datasets and their applications in various research problems. “Artificial Intelligence Applications for Brain MRIS” discusses AI applications in the literature, covering data preprocessing, brain tumor detection, segmentation, classification, survival prediction, tumor and treatment progression, brain modeling, age prediction, gender detection, and brain biometrics. “Materials and Methods” outlines the preferred approaches, used performance evaluation criteria, along with their references, and a summary of the dataset used in this study. In the Benchmark Study Results for Gazi Brains 2020 Dataset chapter, which refers to “Benchmark Study Results For Gazi Brains 2020 Dataset”, deep learning models are designed, developed, applied, and tested on the Gazi Brains 2020 Dataset to achieve tasks such as MRI pre-processing, ROI extraction, anomaly segmentation, and classification-based anomaly detection. The results are also presented using well-known deep learning algorithms and different architectures, along with the XAI results for the first time. “Evaluation and Discussion” provides a general evaluation and discussion on various aspects of the study, with sub-sections including Benchmark Dataset, Test Models based on Deep Learning, Explainability based on XAI Models, Expert Evaluation, and Other Important Issues and Challenges. Finally, “Conclusion and Future Works” concludes the article by summarizing the key objectives, general findings, conclusions, and potential future research directions.

Brain tumor MRI datasets available

Brain tumor benchmarking plays a crucial role in assessing the effectiveness of developed diagnostic methods. However, there are notable discrepancies between existing benchmarks and the available open-access datasets in the literature. Upon reviewing these datasets, it becomes apparent that only a handful are well-structured and meticulously prepared. In contrast, the majority suffer from issues such as poor/inaccurate labeling, incomplete or corrupted data, insufficient quality, and inadequate meta-information, leaving crucial details from previous studies under-explored. These gaps pose challenges to the consistency and reproducibility of research, hindering the broader applicability of these datasets for advancing tumor detection and classification methodologies.

Benchmark datasets are primarily established for specific research areas to characterize behavior using certain standards, such as comparing different techniques, algorithms, architectures, approaches, and models, as well as reducing the time complexity and challenges associated with data collection and various difficult tasks. These datasets also present essential challenges in research areas and support continuous improvement in science. Recently, various health datasets with their contributions and challenges have become publicly available, as shown in Table 1. The purpose of sharing these datasets as benchmarks is to estimate tumors or anomalies, recognize specific brain regions, detect or predict tumor types, and classify or process MRI images, among other tasks. Although these datasets have different advantages, some problems are encountered when using them. Many of the existing databases do not have certain data standards due to using different devices and parameters in acquiring MRI images. Therefore, it is crucial to have more standardized benchmark datasets. Table 1 summarizes the open datasets available in the literature, along with their tumor types, subject information, file types, sequence information, data types, accessibility, and label information.

Table 1 Specifications of brain MRI datasets in the literature.

Name	Tumor types	Subjects	File types	MRI sequences	Data types	Access	Mask/Label	Tesla	
BraTS (Menze et al., 2014)	HGG GBM LGG	2,000 cases 8,000 MRI scans	NIFTI (.nii.gz) DICOM (.dcm)	T1, T1Gd, FLAIR, T2w	Multi-parametric MRI (mpMRI)	Public	1-The necrotic and non-enhancing tumor core (NCR/NET); 2-The perimetral tumor (ED) 4-The GD-enhancing tumor (ET) 0- Everything else	–	
RIDER Neuro (Armato et al., 2008)	Recurrent glioblastoma	19 Patients	DICOM (.dcm)	T1, T2w	MRI	Public	–	1.5 T	
CJDATA (Cheng, 2017)	Meningioma glioma pituitary	233 patients 3,064 images 708 meningioma 1,426 glioma 930 pituitary	MATLAB (.mat)	T1w	MRI	Public	Whole tumor region	–	
CPTAC-GBM (Proteomi, 2018)	Glioblastoma multiforme	66 patients 156,493 images	DICOM (.dcm)	T1, T1c, FLAIR	CR, CT, MRI, SC	Public	–	–	
Acrin-FMISO (Kinahan et al., 2018)	Glioblastoma multiforme	45 patients 670,828 images	DICOM (.dcm)	T1, 3D T2, FLAIR, DCE, DWI/DTI, DSC, MRS	MRI, CT, PET	Public	1-Outer boundary region (Hypoxia mask) 2-Enhancing brain tumor lesions (MR mask)	1.5 T, 3 T	
ACRIN-DSC MR (Kinahan et al., 2019)	Glioblastoma multiforme	123 patients 717,070 images	DICOM (.dcm)	T1w, T2w, FLAIR, DWI, 2D-T1, 3DT1	MRI, CT	Limited	–	1.5 T, 3 T	
QIN-BRAIN DSC (Schmainda et al., 2016)	HGG or GBM, LGG	49 patients	DICOM (.dcm)	T1w	MRI	Public	–	–	
Brain-Tumor Prog (Schmainda & Prah, 2018)	Glioblastoma (newly diagnosed)	20 patients 8,798 images	DICOM (.dcm)	T1w, FLAIR, T2w, ADC	MRI	Public	Whole tumor region	–	
LGG-Deletion (Erickson et al., 2017)	LGG	159 patients 17,519 images	DICOM (.dcm), NIFTI (.nii.gz)	T1w, FLAIR, T2w, ADC	MRI	Public	Whole tumor region	–	
IvyGAP (Shah et al., 2016)	Glioblastoma	39 patients 846,743 images	DICOM (.dcm)	–	MRI, CT	Public	–	–	
QIN GBM treatment response (Mamonov & Kalpathy-Cramer, 2016)	Glioblastoma multiforme	54 patients 589,314 images	DICOM (.dcm)	T1w, T2w, FLAIR, MEMPRAGE, DW-MRI, DCEMRI, DSC-MRI	MRI	Limited	–	3 T	
TCGA-LGG (Pedano et al., 2016)	LGG	199 patients 241,183 images	DICOM (.dcm)	–	MRI, CT, Pathology	Public	–	–	
TCGA-GBM (Scarpace et al., 2016)	Glioblastoma multiforme	262 patients 481,158 images	DICOM (.dcm)	Various	MRI, CT, Pathology, DX	Public	–	–	
REMBRANDT 2011 (Scarpace et al., 2019)	Astrocytoma glioblastoma oligodendroglioma and unidentified tumors	130 patients	DICOM (.dcm)	T1w	MRI	Public	–	–	
REMBRANDT 2018 (Gusev et al., 2018)	Glioma	671 patients	CHP.gz, CEL.gz	–	DNA microarray image	Public	–	–	
SPL brain tumor segmentation db (Surgical Planning Laboratory, 2011)	Meningioma LGG Astrocytoma	10 patients	mrml, xml	T1-spgr	MRI	Public	–	–	
AANLIB (Summers, 2003; Vidoni, 2012)	Glioma, Metastatic Adenocarcinoma,Metastatic Bronchogenic Carcinoma, Meningioma, Sarcoma	Eight patients with tumor three patients for normal	gif	MR-Gad MR-T1 Gad, Spect-Tc, Spect-Tl, MR-T2, MR-PD, CT	MRI, CT, SPECT, Nuclear medicine images	Public	–	–	
BRAINIX (Pixmeo, 2004)	Segmentation	232 Images	DICOM (.dcm)	T1, T1c, T2, FLAIR	MRI	Restricted	–	–	
IBSR (Clearinghouse, 2014)	With Tumors, Normal	21 patients with tumor 21 patients for normal	IMAGE (.img), NIFTI (.nii.gz)	T1w	MRI	Public	Whole tumor region	1.5 T	
Brain tumor connectomics (Aerts et al., 2018)	Glioma (Grade II/III) Meningioma (Grade I/II)	36 patients	NIFTI (.nii.gz)	T1w, DWI, BOLD	MRI	Public	–	–	
Gazi Brains 2020 Dataset (present study) (GaziBrains, 2020)	HGG, normal	50 Normal 50 HGG patients	DICOM for MRI, NIFTI for mask	Flair, T1, T1+C, T2	MRI	Public	12 Labels	1.5 T, 3 T	

The BraTS dataset is currently the most prominent brain tumor dataset available in the literature (Menze et al., 2014). BraTS is updated annually as a challenge dataset, with each iteration incorporating more comprehensive enhancements, and the challenge is repeated accordingly. This dataset includes a segmentation process, as well as survival data. In addition to this dataset, other datasets are available on websites such as The Cancer Imaging Archive (TCIA) (Clark et al., 2013), Harvard SPL (SPL, 2020), OpenNeuro (OpenNeuro, 2021), and other notable public sources. The published datasets were systematically analyzed and compared with each other, as presented in Table 1. The comparison revealed that the datasets were in various file formats, acquired from different types of devices, including various tumor types and MRI sequences, and generally different from each other. For instance, the data available in the literature are mostly for high-grade glioma (HGG) and low-grade glioma (LGG) tumor types, and the most commonly available file formats are Digital Imaging and Communications in Medicine (DICOM) and Neuroimaging Informatics Technology Initiative (NIFTI). The most commonly preferred MRI sequences are T1, T2, FLAIR, and their derivatives, as shown in Table 1.

Although certain differences, such as tumor types, MRI sequences in datasets, and a variety of imaging parameters, may appear acceptable for model development and training, they significantly impact pre-processing stages, establishing architectures, initializing algorithm parameters, establishing model development procedures, and defining other processes. These settings and selections clearly indicate that no specific and standardized brain tumor datasets are available in the literature. As a result, the primary objectives of this study were to prepare the “Gazi Brains 2020 Dataset” as a benchmark dataset and release it for public use (GaziBrains, 2020). As shown in Table 1, the 12 labels within this dataset differentiate it from other datasets. Moreover, the originality of this dataset is emphasized based on various features. Based on the features presented in Table 1, datasets frequently referenced in the literature are preferred according to different specifications and features. Some information about important datasets and general explanations are given below. — BRATS (2021): This dataset comprises images in NIFTI format (.nii.gz) for segmentation and DICOM format (.dcm) for classification. The multiparametric MRI (mpMRI) scans (T1, T1Gd, T2, and T2-FLAIR) are acquired using different clinical protocols and scanners from multiple institutions (Menze et al., 2014).

— RIDER Neuro MRI: This dataset contains MRI images of a total size of 7.3 GB for 19 patients with recurrent glioblastoma (Armato et al., 2008). It includes T1- and T2-weighted images, and the dataset is accessible online (Armato et al., 2008).

— CJDATA (Figshare): Containing 3,064 T1-weighted contrast-enhanced images from 233 patients, this dataset includes three tumor types: meningioma (708 slices), glioma (1,426 slices), and pituitary tumors (930 slices). The dataset is provided in four separate.zip files, each containing 766 slices (Cheng, 2017).

— CPTAC-GBM: This collection originates from the National Cancer Institute’s Clinical Proteomic Tumor Analysis Consortium Glioblastoma Multiforme (CPTAC-GBM) cohort (Proteomi, 2018). Radiology and pathology images from CPTAC patients have been collected and made (Proteomi, 2018) publicly available to investigate cancer phenotypes, which may correlate to corresponding proteomic, genomic, and clinical data.

— Acrin-FMISO Brain: The objective of this dataset is to determine the association of baseline FMISO PET uptake and MRI parameters with overall survival, time to disease progression, and 6-month progression-free survival in participants with newly diagnosed glioblastoma multiforme (GBM). The dataset includes two sets of volumes of interest: enhancing brain tumor lesions and F-FMISO PET hypoxia maps (Kinahan et al., 2018).

— ACRIN-DSC MR-Brain: This dataset consists of MRI and CT images of 123 GBM patients (Kinahan et al., 2019). The MR imaging protocol is divided into two sections: standard and advanced. The standard protocol acquires a pre-contrast T1-weighted, a T2-weighted, a FLAIR, and a diffusion-weighted imaging series, all in the axial plane.

— QIN-BRAIN DSC-MRI: This dataset consists of dynamic susceptibility contrast MRI images of low- and high-grade glioma lesions with binary regions of interest (Schmainda et al., 2016).

— Brain-Tumor-Progression: This dataset (Schmainda & Prah, 2018) includes images from 20 subjects with primary newly diagnosed glioblastoma who were treated with surgery and standard concomitant chemo-radiation therapy (CRT) followed by adjuvant chemotherapy. All images in the dataset are in DICOM format and contain T1w, FLAIR, T2w, ADC, normalized cerebral blood flow, normalized relative cerebral blood volume, standardized relative cerebral blood volume, and binary tumor masks.

— LGG-Deletion: The dataset contains MRIs of 159 subjects with low-grade gliomas (WHO grade II & III) (Erickson et al., 2017). The dataset provides the segmentation of tumors in three axial slices, including the one with the largest tumor diameter and the ones below and above. Tumor grade and histologic type are also available.

— IvyGAP: The Ivy Glioblastoma Atlas Project (Ivy GAP) dataset contains MRI/CT images of brain tumor patients, totaling 846,743 images in DICOM format from 39 patients, including pre-surgery, post-surgery, and follow-up scans (Shah et al., 2016).

— QIN GBM Treatment Response: This collection (Mamonov & Kalpathy-Cramer, 2016) contains “double baseline” multi-parametric MRI images collected on patients with newly diagnosed glioblastoma. The dataset provides clinical image data to establish the test-retest characteristics of parameters calculated from DW-MRI, DCE-MRI, and DSC-MRI, such as ADC, Ktrans, and rCBV.

— TCGA-LGG: This data is a collection of 119 LGG patients. The patients are from Thomas Jefferson University, Henry Ford Hospital, UNC, Case Western, and Case Western St. Joseph’s. The images are available as DICOM files on the website (Pedano et al., 2016).

— REMBRANDT: The Rembrandt dataset includes images of 671 patients collected from 14 contributing institutions between 2004 and 2006. The raw and processed genomics and transcriptomics data are made available via the public NCBI GEO repository. Several updated versions of this dataset also exist (Scarpace et al., 2019; Gusev et al., 2018).

— IBSR: The Internet Brain Segmentation Repository provides manually-guided expert segmentation results and magnetic resonance brain image data (Clearinghouse, 2014). The dataset includes images with a resolution of 1.5 mm.

— Gazi Brains 2020: The newly introduced dataset (GaziBrains, 2020) is briefly explained in the subsection of “The Released Dataset: Gazi Brains 2020 Dataset”. A brief information is given here. Table 2 provides an overview of the limited information on whether the datasets were used for various cases.

Table 2 Comparisons for the datasets in literature according to different use cases.

Available datasets	Tumor segmentation	Tumor classification	Tumor detection	Survival	Tumor treatment Prog.	Brain modelling	MRI preprocess	Age prediction	Gender detection	Other	
BraTS	✓	✗	✗	✓	✗	✗	✗	✗	✗	✗	
RIDER Neuro MRI	✓	✗	✗	✗	✗	✓	✓	✗	✗	✓	
CJDATA figshare	✗	✗	✗	✗	✗	✗	✗	✗	✗	✗	
CPTAC-GBM	✓	✗	✗	✗	✗	✗	✗	✗	✗	✓	
Acrin-FMISO brain	✗	✗	✗	✓	✓	✗	✗	✗	✗	✗	
ACRIN-DSC MR- Brain	✗	✗	✗	✓	✗	✗	✗	✗	✗	✗	
QIN-BRAIN DSC- MRI	✓	✗	✗	✗	✗	✗	✗	✗	✗	✗	
Brain-Tumor- progression	✗	✗	✗	✗	✓	✗	✗	✗	✗	✗	
LGG-1p19qDeletion	✗	✓	✗	✗	✗	✗	✗	✗	✗	✗	
IvyGAP	✗	✗	✗	✓	✓	✗	✗	✗	✗	✗	
QIN GBM treatment response	✗	✗	✗	✗	✓	✗	✗	✗	✗	✗	
TCGA-LGG	✗	✗	✗	✗	✗	✗	✗	✗	✗	✓	
TCGA-GBM	✗	✗	✗	✗	✗	✗	✗	✗	✗	✓	
REMBRANDT 2011	✗	✓	✗	✓	✗	✗	✗	✗	✗	✗	
REMBRANDT 2018	✗	✗	✗	✗	✗	✗	✗	✗	✗	✗	
SPL BTSD	✓	✗	✗	✗	✗	✗	✗	✗	✗	✗	
AANLIB	✓	✓	✗	✗	✗	✗	✗	✗	✗	✗	
BRAINIX	✗	✓	✗	✗	✗	✗	✗	✗	✗	✗	
IBSR	✓	✗	✗	✗	✗	✓	✓	✗	✗	✓	
Brain tumor connectomics data	✗	✗	✗	✗	✗	✓	✗	✗	✗	✓	
Gazi Brains Dataset 2020 (Potential Use-Case)	✓	✓	✓	✓	✗	✗	✗	✓	✓	✓	

Recent advancements in neuro-oncology have been significantly driven by the availability of open access brain tumor imaging datasets. These datasets typically encompass multi-institutional MRI scans representing a range of tumor types—including glioblastomas, meningiomas, and pituitary adenomas—and have enabled the development of deep learning algorithms for tumor segmentation, classification, and outcome prediction. However, as systematically summarized in Table 2, existing resources often exhibit task-specific limitations, rendering them suboptimal for multi-faceted clinical modeling and comprehensive algorithmic benchmarking.

For instance, benchmark datasets such as BraTS, CJDATA, and RIDER Neuro MRI have been widely adopted for tumor segmentation and classification tasks. Yet, they generally lack support for broader clinical objectives such as survival prediction, treatment progression modeling, or MRI progression tracking. Moreover, a majority of these datasets do not include demographic metadata (e.g., age, sex), precluding analyses related to population diversity, bias assessment, and personalized modeling. Others, such as QIN-BRAIN DSC-MRI, provide specialized imaging for select tumor characteristics but suffer from narrow scope, reduced generalizability, and limited annotation coverage. Additionally, only a small subset of datasets enable integrative modeling across multiple domains (e.g., imaging, clinical metadata, progression metrics), which limits the development of generalized, clinically translatable AI systems. This fragmentation poses a significant barrier to reproducibility, cross-study comparison, and the development of end-to-end diagnostic pipelines.

Table 2 consolidates the comparative analysis by offering a structured overview of the current landscape of publicly available brain tumor datasets, elucidating key technical trends and limitations that hinder the development of generalizable AI solutions in neuro-oncology. As reflected in the table, the majority of benchmark datasets—despite their contributions—are predominantly constrained to singlepurpose tasks such as tumor segmentation or classification. A much smaller subset extends to more advanced clinical tasks like survival prediction or treatment response modeling, and only a few support integrative applications such as MRI progression tracking or demographic inference (e.g., age and gender prediction). This segmentation of capabilities results in fragmented workflows, impeding the creation of unified, multi-objective models and consistent cross-dataset evaluations.

Moreover, Table 2 underscores the lack of modality uniformity across datasets, with many relying on limited MRI sequences (e.g., only T1 or T2-weighted images), thereby restricting the scope of feature extraction and clinical utility. Likewise, demographic metadata—crucial for personalized medicine and bias mitigation—is absent or sparsely represented in most datasets, further diminishing their translational relevance.

Against this backdrop, the Gazi Brains 2020 dataset presents a strategically curated resource designed to address these precise shortcomings. As delineated in Table 2, it is one of the most comprehensive datasets currently available, providing multi-parametric MRI data alongside detailed annotations for a wide array of downstream tasks. Specifically, it supports: Tumor segmentation, classification, and detection,

Demographic inference tasks such as age and gender prediction.

Notably, Gazi Brains 2020 is among the very few datasets that integrate multi-task labeling, demographic metadata, and modality diversity within a unified framework. This enables the development of more robust, generalizable models that can be evaluated across multiple clinically relevant tasks using a consistent data source. By integrating this multi-task design within a standardized annotation and preprocessing framework, Gazi Brains 2020 not only addresses key deficiencies in existing resources but also sets a precedent for future dataset development.

Furthermore, the dataset mitigates prevalent challenges in the literature—such as annotation inconsistency, class imbalance, and missing metadata—by employing standardized preprocessing pipelines and a curated labeling schema. Its design supports both supervised and semi-supervised learning paradigms, making it suitable for a wide range of algorithmic approaches, including transfer learning and federated learning models.

In addition to its multimodal architecture and extensive annotation schema, the Gazi Brains 2020 dataset includes unique label types that directly enable research tasks typically unsupported by other benchmark datasets. For example, the availability of demographic metadata—such as age and gender—not only facilitates demographic inference tasks (e.g., age estimation, and gender classification) but also supports bias assessment and subgroup analysis, which are crucial for developing equitable AI systems. Moreover, the inclusion of a wide array of tumor types and grades enhances the ability to train models that generalize across both common and rare pathologies, reducing the impact of class imbalance. The consistent presence of multiple MRI modalities (e.g., T1, T2, FLAIR) across subjects also supports systematic modality ablation studies, allowing researchers to evaluate the individual and collective diagnostic value of each imaging sequence. Collectively, these features position Gazi Brains 2020 as a uniquely versatile dataset that bridges technical gaps in the existing landscape and enables a broader spectrum of clinically meaningful research applications.

In summary, Gazi Brains 2020 is positioned not merely as a complementary dataset, but as a foundational benchmark that expands the scope of existing resources. By enabling multifactorial analysis across imaging, clinical, and demographic domains, it offers a scalable, reproducible, and clinically meaningful platform for advancing research in brain tumor diagnostics.

The application domains and downstream tasks enabled by the Gazi Brains 2020 dataset are further elaborated in “Artificial Intelligence Applications for Brain MRIS”, with supporting references from relevant literature. Preliminary benchmark studies (GaziBrains, 2020) demonstrate that the dataset exhibits high-quality annotations and robust segmentation accuracy across various tumor types. These characteristics render it particularly suitable for a wide range of AI-driven applications, including—but not limited to—tumor segmentation, classification, detection, survival prediction, anomaly detection, and demographic attribute estimation (e.g., age and gender), as systematically summarized in Table 2.

Moreover, when cross-referenced with the technical specifications provided in Table 1, the Gazi Brains 2020 dataset emerges as a uniquely versatile resource, encompassing the multimodal, multi-task, and metadata-rich attributes required for developing clinically generalizable models. A detailed account of these dataset characteristics, including imaging modalities, annotation protocols, and task-specific configurations, is provided in “Materials and Methods”.

Artificial intelligence applications for brain MRIs

Studies in the area of brain tumor detection and classification have been conducted in the literature using AI-driven solutions on machine learning/deep learning-based models. Machine learning-based models such as ANN, support vector machine (SVM), and K-nearest neighbors need to prepare manual feature extraction. Deep learning-based models can automatically extract features from the MRIs and have been shown to perform the tasks more accurately in many cases. Deep neural networks (DNNs) have been widely used in brain tumor detection and classification tasks, such as convolutional neural networks (CNNs) and their various versions, generative adversarial networks (GAN), etc.

Brain tumor image datasets have been used for segmentation, tumor detection, classification, survival prediction, treatment, and tumor progression. All of these help us to better understand tumors’ type, location, size, progression, growth, and spread, as well as to know the brain’s structure of functions and to improve diagnosis or treatment more and more. It is also used in different areas such as modeling and image operations, pre-processing steps, etc. This section summarizes the usage areas of brain tumor image datasets; the studies carried out in these areas are summarized; and the benchmark datasets and their usage in the literature are demonstrated in Table 2.

Since brain tumors can be very dangerous, early detection and grading are vital. The diagnostic process consists of three key procedures: detection, segmentation, and classification (Abd-Ellah et al., 2019). Initially, the presence of a tumor is identified, followed by the segmentation of the affected region. Subsequently, classification is performed to determine whether the tumor is benign or malignant, and its grade is assigned (Grades I–IV) (BrainTumorBasics, 2024). Although these processes—detection, segmentation, and classification—may appear distinct, they are inherently interconnected and collectively contribute to accurate diagnosis and treatment planning.

The diagnosis process is performed traditionally by experts, as well as by computer-aided design systems (CADs), which can be used as decision support systems. This study focuses on computer-assisted systems that use traditional machine learning and deep learning-based systems (Abd-Ellah et al., 2019). In addition, when classical machine and deep learning methods are used alone, bottlenecks occur after the performance rate reaches a certain level. At this point, hybrid methods are very useful in overcoming these bottlenecks. This section evaluates different algorithms applied to medical problems using brain tumor MRI datasets. According to the literature, the study topics and problems related to brain tumors can be classified and explained in the following subsections.

Brain tumor detection

Tumor detection can be defined as the first step for brain tumor diagnosis. Various machine learning studies have been conducted across different subjects (Ghosh & Kole, 2021; Hussain et al., 2019; Saeedi et al., 2023; Abdel-Maksoud, Elmogy & Al-Awadi, 2015; Nayak et al., 2018). Such studies provide the opportunity to see that artificial intelligence approaches have been used in different areas, regardless of the problem, and to use the developed methods in different areas. Nayak & Kengeri Anjanappa (2023) used the naive Bayes classification method for brain tumor detection. Abd-Ellah et al. (2018) used AlexNet and Virtual Geometry Group (VGG) 16 and 19 for detection and localization. In the study of Shakeel et al. (2019) a machine learning-based back propagation neural network (MLBPNN) was analyzed with the help of infrared sensor imaging technology. The features were extracted using the fractal dimension algorithm. Thus, the results were obtained using AdaBoost and MLBPNN classifiers. Ozyurt, Sert & Avci (2020) used the fuzzy C-means with super-resolution and CNN with extreme learning machine algorithms for brain tumor detection. Some recent studies on brain tumor detection are also presented here.

Sadad et al. (2021) developed a deep learning model that can detect and segment different types of tumors with a ResNet50 backboned U-Net architecture. In the preprocessing step, high-resolution images were obtained with the contrast by stretching algorithm, and data augmentation was made with horizontal and vertical flips (Sadad et al., 2021). Islam et al. (2021) developed a brain tumor detection approach using super-pixels, template-based k-means (TK), and PCA algorithms with low computational requirements. Mean and median filters were used for image enhancement and noise reduction. Feature extraction was performed using super-pixels and PCA, and detection was performed with the TK algorithm (Islam et al., 2021). Arif et al. (2022) developed a tumor detection approach that consists of preprocessing, segmentation, feature extraction and selection, and SVM classification steps. Thresholding, morphological operation, and region filling were used in the preprocessing step. Then segmentation was performed using Berkeley Wavelet Transformation. On the obtained outputs, feature extraction was performed with GLCM, and feature selection was performed with the genetic algorithm (GA). Finally, the Visual Bag-based SVM was used for the classification task.

Brain tumor segmentation

Segmentation refers to separating the image data into ROIs to reveal the characterization of the data and to facilitate the detection of its content and visualization (Abd-Ellah et al., 2019). In brain tumor segmentation, methodologies can be broadly categorized into traditional machine learning methods and next-generation deep learning methods. First, the studies used in machine learning are analyzed. Tahir et al. (2019) studied pretreatment techniques for segmentation and classification. Kumar, Krishna & Kusumavathi (2019) used the GA for feature selection. Faragallah, El-Hoseny & El-sayed (2023) used the k-means algorithm for automatic segmentation. Ahmadvand, Daliri & Zahiri (2018) used a dynamic classifier selection Markov random field for supervised segmentation. Zhao et al. (2019) used the fuzzy clustering algorithm for noisy image segmentation. Al-Dmour & Al-Ani (2018) used an artificial neural network (ANN) for brain MR tissue segmentation. In another study, Wang, Cheng & Basu (2010) performed a fully automatic brain tumor segmentation using the gaussian bayesian classifier. Pohl et al. (2002) used the expectation-maximization algorithm for segmentation. Srinivas & Rao (2018) evaluated their method with parameter calculation after they used fuzzy c-means and k-means clustering algorithms.

Secondly, studies using deep learning methods for brain tumor segmentation tasks have been extensively investigated. Van Opbroek et al. (2015, 2018) applied transfer learning for image segmentation. Additionally, studies have been conducted in which segmentation and classification were performed using Faster R-CNN by Kaldera, Gunasekara & Dissanayake (2019a, 2019b). In another study, Gu & Tresp (2020) conducted studies to increase the performance of capsule networks (CapsNets), which give better results than CNN. Studies using deep convolutional encoders and decoders are also available (Dheepa & Chithra, 2023; Afshar et al., 2018b). Xiao et al. (2016) used a stacked denoising auto-encoder for brain tumor segmentation. Mlynarski et al. (2019) used 2D CNNs for feature extraction and 3D CNN for segmentation in multisequence MRI. Kamnitsas et al. (2016) used the deep learning-based DeepMedic method, which consists of 11 layers of multi-scale 3D CNN for segmentation. Wang et al. (2019b) used 2D MRI slices from 3D MRI and normalized images and achieved segmentation by the WRN-PPNET model (wide residual and pyramid pool network) for fully automatic brain tumor segmentation. Hussain, Anwar & Majid (2018) preprocessed the data and divided it into patches, which are passed through a deep CNN to predict the output labels for individual patches for brain glioma tumor segmentation. Zhao et al. (2018) used FCNNs and conditional random fields (CRF) for training and applied fine-tuning using images. Pereira et al. (2017) used FCNN for hierarchical brain tumor segmentation. For this purpose, firstly, the whole tumor is segmented, and then intra-tumor tissue identification is done.

Hybrid methods are also used in segmentation. Rao & Lingappa (2019) used kernel-based fuzzy c-means clustering and CNN as hybrid (Hybrid KFCM-CNN). Mittal et al. (2019) used stationary wavelet transform (SWT) and growing convolutional neural network (GCNN) together. The GCNN is part of the method that automates the process. These two studies serve as examples of how machine learning and deep learning are used together. Sajid, Hussain & Sarwar (2019) used a hybrid CNN method that uses a patch-based approach and considers both local and contextual information while predicting. In the study of Anand Kumar & Sridevi (2018), non-uniformity normalization was used in the preprocessing step; then a gray-level co-occurrence matrix was used for feature extraction, and 3D CNN automatically segmented the tumors. Ahmad et al. (2019) used a 3D dense dilated hierarchical model. Ibtehaz & Rahman (2020) (as MultiResUnet) and Li, Li & Wang (2019) used a modified version of U-Net, one of the most popular deep learning architectures for image segmentation. Gu et al. (2019) proposed a context encoder network (CE-net) for image segmentation and compared their study with the classical U-Net metho. They used the ResNet block as the feature extractor in this study. Peng et al. (2020) benefited from multiscale 3D U-Nets architecture that uses several U-Net blocks. Dolz et al. (2018) used HyperDenseNet, a 3D fully CNN approach for multi-modal brain image segmentation. Casamitjana et al. (2017) used a V-Net approach using ROI masks. In addition to machine learning and deep learning, supervoxel-based segmentation methods, which enter the computer vision field, are also used (Huang et al., 2018; Yang et al., 2017). Neuro-oncology studies also use brain tumor MRI datasets (Schmainda et al., 2018). Other recent studies are summarized as follows. Ranjbarzadeh et al. (2021) proposed the Cascade CNN (C-CNN) approach, a flexible and effective solution for the segmentation task. A new Distance-Wise Attention mechanism was developed to take into account the central location of the tumor within the model. In the preprocessing step, z-score normalization, thresholding on various measurements, and expected tumor area calculations were made. With the proposed method, a more effective approach was obtained by processing the relevant regions rather than the whole image (Ranjbarzadeh et al., 2021). Zhou et al. (2021) developed a new model for segmentation named efficient 3D residual neural network (ERV-Net). A fusion of Dice and cross-entropy losses was used to assist the model convergence and the dataset imbalance problem. Before the model training, data augmentation methods such as Gamma correction, random crop, Gaussian noise, and random elastic deformations were applied. A post-processing algorithm based on neural network (NN) characteristics and tumor distributions was developed and implemented to improve performance. The proposed method has achieved superior success with its low computational cost and high performance (Zhou et al., 2021). Wang et al. (2021) developed an encoder-decoder-based TransBTS NN model for segmentation. 3D CNN and transformers were used for feature extraction in the encoder part, and up-sampling was used to estimate the segmentation maps in the decoder part. The proposed approach has achieved comparable or superior performance on various datasets in 3D segmentation.

Brain tumor classification

When classification studies using machine learning methods are evaluated, Tripathi & Bag (2020) performed tumor grading using random forest, SVM, and decision trees. A similar study was performed by Mitra, Tripathi & Bag (2020). Tahir et al. (2019) also used machine learning. Iqbal et al. (2018) used binary classification. Kumar, Krishna & Kusumavathi (2019) benefited from the genetic algorithm while segmentation and classification. Ismael & Abdel-Qader (2018) used the backpropagation algorithm and statistical features. It is vital to develop effective methods not only in the brain tumor classification problem, but also in other classification problems where feature selection is much more meaningful (Ozcelik & Altan, 2023). The developed models provide different perspectives for different classification problems. In another study, Ayadi et al. (2019) benefited from Discrete Wavelet Transform (DWT) and Bag-of-Words (BoW). Zia et al. (2017) used nonsubsampled contourlet transform (NSCT) and isotropic gray level co-occurrence matrix (GLCM) as a pretreatment for feature extraction, while they used SVM-based method (Zia et al., 2017) when classifying three grades of glioma (Grades II, III and IV) like (Chandra & Bajpai, 2020). Srinivasan & Nandhitha (2019) used GLCM and Wavelet Transform. Al-Zurfi, Meziane & Aspin (2019) followed the path for brain glioma tumor diagnosis: preprocessing, then MRI Segmented ROI, feature extraction with 2D and 3D GLCM, feature selection, and classification.

Brain tumor classification studies using deep learning methods are well-studied problems in the literature. CNNs have been preferably used in the classification (Al-Zoghby et al., 2023; Ucuzal, Yasar & Colak, 2019; Kotia, Kotwal & Bharti, 2019; Shaikh, Kollerathu & Krishnamurthi, 2019; Kumar & Kumar, 2023; Ozkaraca et al., 2023; Deepak & Ameer, 2019). In addition, there are studies in which CNN is combined with other methods. For example, Afshar, Mohammadi & Plataniotis (2018a) used capsule networks (CapsNets) with CNN. Ozyurt et al. (2019) used neutrosophy and CNN (NS-CNN). In their method, MRI images were segmented using neutrosophic and expert maximum fuzzy-sure entropy (NS-EMFSE) approach. The features of the segmented brain images in the classification stage were obtained by CNN and classified using SVM and KNN classifiers. There are also studies in which CNN and extreme learning were used together by Pashaei, Sajedi & Jazayeri (2018), Pashaei, Ghatee & Sajedi (2020). Deepak & Ameer (2019) used the features of CNN with transfer learning. Apart from these, Sajjad et al. (2019) used extensive data augmentation with CNN. Talo et al. (2019) studied brain abnormality classification. They used CNN-based ResNet34 and benefited from transfer learning, data augmentation, optimal learning rate finder, and fine-tuning. Murali & Meena (2019) used R-CNN, and Kaldera, Gunasekara & Dissanayake (2019a) used faster R-CNN for classification. In another interesting study, Ghassemi, Shoeibi & Rouhani (2020) used generative adversarial networks (GAN). Swati et al. (2019) used transfer learning and fine-tuning. Another study that utilizes transfer learning was the research conducted by Rehman et al. (2020). Methods using CapsNets in the literature are also quite high (Afshar, Mohammadi & Plataniotis, 2018a; Vimal Kurup, Sowmya & Soman, 2019; Adu et al., 2019). Zhou et al. (2018) used DenseNet and recurrent neural network (RNN). Studies are also conducted with ResNet (Ismael, Mohammed & Hefny, 2020; Sharma et al., 2023; Mehnatkesh et al., 2023). Cheng et al. (2019) used Convolutional CapsNets and produced the ConvCaps concept.

Hybrid methods used for classification other than those mentioned above were examined. Cogan, Cogan & Tamil (2019) used SVM for features and ResNet-101 and Faster R-CNN for detection. In a study, Kutlu & Avci (2019) used CNN, DWT, and long short-term memory (LSTM) together. Anaraki, Ayati & Kazemi (2019) focused on the classification problem using CNN and genetic algorithm. Studies on tumor classification have been extensively explored. In particular, different studies are carried out to develop approaches that increase performance. It is understood that studies focus on different models related to CNN. Ayadi et al. (2021) developed a new CNN model for classifying different types of tumors using various datasets. Small-sized kernels and strides were used in the CNN model. Data augmentation was used to increase performance. Raza et al. (2022) developed a hybrid approach called DeepTumorNet for glioma, meningioma, and pituitary tumor classification. The last five layers of GoogLeNet were discarded, and 15 new layers were added. In the preprocessing stage, normalization was performed on the images. The model achieved superior performance compared to similar studies (Raza et al., 2022). Sharif et al. (2022) developed a tumor classification approach that includes CNN feature extraction, feature selection, and SVM classification steps. Feature selection was made on the features extracted by the DensetNet201 model using the Entropy–Kurtosis-based High Feature Values (EKbHFV), which is a new approach, and modified genetic algorithm (MGA), and these features were fused. Finally, tumors were classified with the cubic SVM. Choosing the optimum features and reducing the classification time improved the performance of the model.

Survival prediction

The other subject of this article is survival prediction. The basic logic here is to estimate whether the patient will die or how long the patient might survive according to the datasets of MRI images (Pérez-Beteta et al., 2018; Boxerman et al., 2016; Ratai et al., 2018). In the study by Pérez-Beteta et al. (2018) the patient’s response to the operation was monitored, and it was estimated whether he would survive or not. Kaushik, Kumar & Rashmi (2019) studied the prediction of survival time of brain tumor patients using a denoising wavelet transform and SVM. Liu et al. (2016) used transfer learning (ImageNet) to predict survival. Ahmed et al. (2017) also used pre-trained CNN (ImageNet ILSVRC) by fine-tuning a small dataset for the same subject. Nie et al. (2019) used CNN and multi-channel CNN architecture to train survival prediction models and also used an SVM classifier. It is understood that after increasing the consistent data in this area, different studies that contributed to the literature were carried out. Some recent work is summarized here.

Ammari et al. (2021) used radiomic signatures and patient age information to estimate overall survival time in months for glioblastoma (GBM) patients using K-NN, RF, logistic regression gradient boosting, AdaBoost, naive Bayes, and SVM machine learning models. Image quality was improved by using advanced normalization tools (ANTs) and HD-BET brain extraction tools in the preprocessing step. Similarly, Das et al. (2022) used algorithms such as RF, SVM, and XGBoost to estimate the survival time of GBM patients. Before the estimation process, tumor regions were segmented with the U-Net++ model, and feature selection was made with PCA. GA and PSO algorithms were applied to the fused features for better performance, and the highest performance was obtained with the SVM classifier (Das et al., 2022). In another study, Islam, Wijethilake & Ren (2021) developed an FCN and Conditional GAN (cGAN)-based model to complement the missing MRI sequences. This model has also been used for tumor segmentation. The proposed model includes octave convolutions and a new decoder architecture, skip-scSE. Radiomic feature extraction was applied to the segmentation outputs, and feature selection was made by recursive feature elimination. Then, survival time was estimated with the regression model. It was concluded that the completion of the missing MRI sequences greatly contributed to the performance.

Tumor treatment progression

The tumor expansion change is to follow the progress of the tumor over time. In this way, information is obtained as to whether surgical intervention is performed on the tumor and how the patient’s condition is. Treatment progress is to monitor how the patient reacts to the treatments and drugs applied to the patient. There are many studies on tumor and treatment monitoring using the relevant datasets. Thanks to these studies, the results of the treatment applied to the patient can be monitored, and the tumor development can be followed. Tumor progression was observed in a study by Pérez-Beteta et al. (2018). In a study by Kettelkamp & Lingala (2020) a new patient-specific treatment method was developed, and brain tumor progression was observed. Galldiks et al. (2017) performed the treatment with brain tumor monitoring. Fernandes et al. (2020) developed a helpful tool for early treatment planning. The developed tool provides four basic findings together with preprocessing, segmentation, shape, texture feature extraction, and classification steps and provides helpful decision support to doctors/experts for treatment planning.

Brain modeling

Brain modeling involves simulating the chemical and electrical properties of neurons. Brain tumor image datasets have been used quite widely for different modeling and purposes. In the studies, Zhan examined the brain tumor through a mathematical model and brain model geometry (Zhan & Wang, 2018; Zhan, 2020). Beers et al. (2018) studied pharmacokinetic modeling in their research. Shen et al. (2019) modeled connectome-based brain modeling on brain tumor connectomics data. Amico et al. (2019) used the same dataset for modeling communication dynamics. Aerts et al. (2018) studied modeling brain dynamics. In another study, Kaboodvand (2019) worked on brain connectivity and dynamics modeling. Kamath, Rudresh & Seelamantula (2019) made modeling of Fourier descriptors. Eyles developed a tractable model for tumor growth (Eyles, 2019). In another study, Kalloch et al. (2019) studied the simulation of the interaction of the human body. Current studies also show that studies including visualization have been carried out. Aerts et al. (2020) modeled brain dynamics after tumor resection using the Virtual Brain tool. The parameters before and after the resection were examined and compared with the bases obtained from healthy subjects. In light of these inferences, modeling was achieved by performing a virtual resection of a patient with a brain tumor. Li & Yap (2022) conducted a review on the application of generative modeling on MRI images in another brain modeling study. It has been concluded that traditional modeling is insufficient; the use of generative models is more effective; and the transition from descriptive connectome to mechanistic connectome makes it open to innovations.

MRI preprocessing

In medical studies, where different studies are carried out, very different outputs are obtained according to the different characteristics of the devices from which the images are obtained. For example, raw data on the brain obtained by MRI may not always be suitable for direct use. Therefore, preprocessing steps are needed. Since the datasets in this field of interest are sometimes pixel-based picture data or voxel-based tensor data, they are in the field of interest, such as image classification, image cropping, image processing, neuro-imaging, and computer vision, as well as topics such as tumor classification. In this section, various studies on preprocessing in making datasets ready for use are briefly reviewed.

Phaye et al. (2018) proposed a framework that uses CapsNets for image classification and object recognition. Young (2016) used spatial pyramid match kernels for brain image classification. Zia, Akhtar & Aziz (2018) studied image cropping. Afshar et al. (2018b) used autoencoders for inter-slice interpolation of brain tumor volumetric images. Anwar, Arshad & Majid (2017) developed a medical image retrieval system. Ou et al. (2018) worked on view normalization. There are also studies on Lossless Image Compression (Sharma, Sood & Puthooran, 2020, 2021). Bejinariu et al. (2015, 2014) worked on image processing and image registration.

Tolstokulakov et al. (2020) investigated the effect of multi-channel input MR images on deep learning models. For each slice in the T1, T1C, and FLAIR sequences, a new RGB image was obtained by summing the other two slices adjacent to that slice. In addition, an RGB image was obtained by combining the T1, T1C, and FLAIR sequences of the same slice level. The image enhancement processes increased the segmentation performance. Similarly, Groza et al. (2020) used the combinations of T1, T1C, and FLAIR slices to improve segmentation performance. Experiments were done by taking weighted combinations of T1, T1C, and FLAIR slices in a single channel and distributing them among RGB channels. Thanks to the preprocessing step, the segmentation performance increased. Maurya & Wadhwani (2022) used an anisotropic diffusion filter (ADF) for noise reduction, skull stripping, and contrast enhancement preprocessing steps for better image quality to improve detection and segmentation performance. They achieved better results in terms of computational cost and PSNR.

Age prediction

Research on the brain is extensively studied in the literature. Previous research showed the relationships between brain development and the ages of individuals, yet there are limited studies examining the correlations among brain vs. age, social class, race, and region (Taki et al., 2004; Smith et al., 2007; Philippe & Davison, 1996). In addition to these subjects, the effects of annual changes on brain development are also examined in the literature (Resnick et al., 2000). Dosenbach et al. (2010) estimated brain maturity using fMRI with SVM. Another study looked at the relationship between chronological and estimated ages (Jónsson et al., 2019). The literature review has shown that the age prediction problem is still there and requires more data to be determined.

With the recent publication of different datasets in the relevant literature, it is clear that studies that can be considered effective have been revealed. A study in 2021 (Asan, Terzi & Azginoglu, 2021) emphasized the importance of age estimation in the early diagnosis of Alzheimer’s and Parkinson’s diseases. In addition, it was stated that a three-dimensional structure should be used when brain MRI and age information were considered together for each person in the dataset, and the 3D convolutional neural network (3D-CNN) approach was preferred. Thus, the importance of choosing three-dimensional deep learning approaches for predictions on specific subjects such as age was expressed. In the related study (Asan, Terzi & Azginoglu, 2021), the importance of using datasets consisting of anomaly data in relation to age estimation to improve performance was emphasized. Considering the dealing with similar studies in the literature, it is stated that increasing the number of datasets proposed in this study enables the conduct of research offering diverse perspectives.

Gender detection

Just like examining age estimation on medical data, research studies on gender prediction are also important topics. Inferences about investigating the relationship between the brain and gender are critical for general predictions. Studies with gender determination are available in the literature for the datasets of Alzheimer (Long & Holder, 2012), Brain Connectivity (Sen & Parhi, 2019), and Human Connectome (Gao et al., 2019; Hu, Luo & Zhao, 2019). Smith et al. (2007) examined the effects of age and gender on the human brain anatomy. Similar to age estimation, gender detection is observed to be challenging to perform. One of the main problems in this context is that the datasets are not designed specifically to solve these problems.

Despite the different problems, studies have been focusing on this subject recently. For instance, Wahlang et al. (2022) focused on both age estimation discussed in the previous section and age estimation based on medical data expressed in this section. While analyzing the test process of the study, it was stated that effective classification can be made using some CNN-based architectures. In addition, the authors stated that it is more beneficial to use preliminary information on age and gender. This information plays a key role in classification and states that different brain tumor analysis approaches are considered basic factors. It is planned to focus on gender prediction within the scope of future studies when a sufficiently comprehensive dataset is reached on this subject (Wahlang et al., 2022).

Brain biometrics

Traditional biometrics, including fingerprint, facial recognition, iris scanning, voice recognition, and DNA analysis, have been thoroughly examined in scholarly literature and broadly implemented in practical applications. Nonetheless, each of these biometrics has its vulnerabilities (Jain, Ross & Prabhakar, 2004). Thus, a new biometric trait more secure than conventional biometrics should satisfy two criteria: it would be more challenging to steal and cancelable. Despite the possibility of distinct brains exhibiting identical features and shared qualities, scientists have determined that no two brains are or will ever be identical (Gage & Muotri, 2012). The evolution of the human brain is influenced by genes, inheritance, experiences, and the lessons we take away from them. This makes the human brain exceptional in its powers and design, almost appearing to be the product of extraordinary brilliance. Although the brain’s microstructure may not be stable at this early stage of development, its macrostructure is completely stable (Chris Fraley, 2002). In addition to the structural feature of the brain, functional network connectivity has been recognized as a method to characterize brain activities, serving as a form of “brain fingerprinting” to distinguish a person from a group of people. These structural and/or functional features of the brain measured may meet the biometric traits criteria mentioned above.

Research in brain biometrics mostly involves the extraction of structural and/or functional characteristics of the brain and the evolution of matching or classification algorithms by using these characteristics (Bhatnagar & Mishra, 2020; Zhang et al., 2023). The researchers employed several statistical analyses, conventional machine learning, and deep learning techniques to enhance matching or classification performance. Substantial individual variations in brain activation have been the focus of mainstream fMRI research (Cai et al., 2021; Sarar, Rao & Liu, 2021; Lori et al., 2018; Chen & Hu, 2018; Hassanzadeh & Calhoun, 2020; Wang et al., 2019a). Identifying/verifying subjects from a large group has been effectively accomplished through individual heterogeneity in functional connectivity (FC). Brain biometrics using different deep learning models such as autoencoder (Cai et al., 2021), shallow feedforward neural networks (Sarar, Rao & Liu, 2021), recurrent neural networks (Chen & Hu, 2018), and deep siamese networks (Hassanzadeh & Calhoun, 2020) have been carried out using functional connectivity data from fMRI.

Other subjects

When the literature is examined, it is seen that brain tumor datasets are used in many different areas apart from the ones given above. For example, Fernando et al. (2019) detected anomaly using neural memory networks. There are other examples, like expression studies for glioblastoma (Pérez-Beteta et al., 2018; Close et al., 2020). Satriadi et al. (2017) studied 3D visualization. Ocegueda et al. (2016) studied the computation of integrals in mathematics. Prokopenko et al. (2019) studied synthetic CT image generation from MRI scans using GAN. There are also studies on connecting cancer phenotypes to genotypes (Grossmann et al., 2016; Silva et al., 2016). In addition, extensive studies on the relationship between brain pathologies and segmentation, as well as medical diagnosis, surgical planning, and disease development, have been presented to the literature (Havaei et al., 2016). Another field of study is neuroimaging and imaging in neuro-oncology (Mankoff et al., 2017; Sonni et al., 2018; Wolf, 2019). In one study (Paquola et al., 2021), a tool called BigBrainWarp was proposed in integration with Neuroimaging. Here, 3D imaging technology was used. Thus, potentials that offer multi-scale investigations of brain organization were mentioned. In the other study presented by Kim et al. (2021) systematic research and meta-analysis were performed to evaluate the incidence of neurological complications and detailed neuroimaging findings based on images associated with coronavirus disease (COVID-19) using MRI. When the outcomes were evaluated, it was reported that abnormal neuroimaging findings were occasionally observed in COVID-19 patients. It has been stated that critically ill patients showed abnormal neuroimaging findings more frequently than other group members (Kim et al., 2021). Research studies on the brain have been conducted in many different areas, and many different studies have been carried out in the evaluation of the quality of life. The study presented by Pereira-Sanchez & Castellanos (2021) conducted significant research work on neuroimaging on attention-deficit/hyperactivity disorder (ADHD). Inferences were presented regarding the potential of promising studies on the neurobiology of ADHD.

Materials and Methods

This chapter has presented a comprehensive overview of the dataset, the models employed, the experimental setup, and the evaluation metrics utilized in this study. It outlines the processes followed, the progress made, and the tests conducted throughout the research. A detailed dataset analysis has been provided, including its sources, preprocessing steps, and key attributes. The models have been discussed in terms of their architecture, parameters, and the rationale for their selection. Furthermore, the evaluation metrics have been extensively explained to underscore their significance in assessing model performance.

The released dataset: Gazi Brains 2020 dataset

The presented dataset (GaziBrains, 2020) in the project of TBP (https://cbddo.gov.tr/en/projects/turkish-brain-project) consists of glioma-type brain tumors prepared by six medical experts from the Faculty of Medicine at Gazi University. Gazi Brains 2020 Dataset includes brain MR images of 100 patients, 50 of which were healthy and 50 of which were High-Grade Glioma (HGG). This dataset includes T1-weighted, T2-weighted, FLAIR, and T1-weighted (T1+C) MRI sequences for all patients. These data were obtained in the 2018–2019 period, and the dataset contains HGG patients with the segmentation masks. All MRI images were segmented by medical experts in the project. In addition, the dataset includes tumor region segmentation and 12 anatomical structure tags prepared. Gazi University Clinical Research Ethics Board granted Ethical approval to conduct this study within its facilities by Decision No. 616.

While preparing the Gazi Brains dataset, the real-life difficulties encountered during the diagnosis made on MRI images were taken into consideration. In this context, a dataset with various patient demographic features was obtained from different institutions and MRI devices belonging to different brands with different qualities. Data for 50 healthy and 50 patients with HGG were collected to have a balanced dataset. The dataset contains different sequence types such as FLAIR, T1-weighted, and T2-weighted for normal patients, and their sequence counts are 1,016, 1,008, and 1,008 images, respectively. Moreover, there are 1,062, 1,054, 1,057, and 1,056 images from patients for FLAIR, T1-weighted, T1+C, and T2-weighted sequences. The dataset includes tumor findings and components of HGG patients and demographic characteristics such as age and gender. MRI studies in the datasets were acquired from 24 different institutions nationwide with four different brands (masked with the label ‘unique’) of MRI scanners with a magnetic field strength of either 1.5 or 3 Tesla. General information about the dataset, including statistical features, is presented in Fig. 1. In Figs. 1A and 1B, the age of the patients in the dataset (the average age for normal patients is 38.7, while the average age for patients with HGG is 56.8) and patient gender distributions are given. In Fig. 1C, the institutions are masked with a unique label for privacy concerns. In Figs. 1D, 1E, and 1F, the tesla values, slice thickness, and brands of the MRI device are given according to normal and HGG patients, respectively.

Figure 1 General information for Gazi Brains 2020 dataset.

(A) Age distribution. (B) Gender distribution. (C) MRI Institutions. (D) MRI slice tesla distribution. (E) MRI slice thickness. (F) MRI brands.

The data obtained after all processes were anonymized in accordance with data confidentiality and public sharing criteria. MRI studies for all subjects include T1-weighted, T2-weighted, fluid attenuation inversion recovery (FLAIR) sequence images all in the axial plane. The information of Axial T2-weighted (A), T1-weighted (B), FLAIR sequence (C), and post-contrast T1-weighted (D) images with tumor segmentation on post-contrast T1-weighted images (E) of a high-grade glioma patient is presented in Fig. 2. Sequential segmented post-contrast T1-weighted MRI images of a patient with high-grade glioma are given in Fig. 3. The outputs presented step-by-step based on these images are essential for following basic operational processes. Each image in Fig. 3 shows the process changing from left to right and top to bottom.

Figure 2 A sample patient with HGG according to different perspectives.

Figure 3 T1-weighted MRI images after segmentation procedures.

MRI studies of all HGG patients and 12 normal subjects include additional axial plane post-contrast (Gadolinium) T1-weighted images. All MRI studies were available in Digital Imaging and Communications in Medicine (DICOM) file format. This dataset was compiled from 50 normal subjects and 50 histologically proven high-grade glioma (HGG) patients, with 12 labels of anatomical structures, non-tumoral findings, and tumor components, which are cross-validated among six medical experts. There are eight different labels for anatomical structures and non-tumoral findings, which are as follows: region of interest (includes all other labels), eyes, optic nerves, lateral ventricles, third ventricle, ischemic gliotic changes, cavum septum pellucidum, and intracerebral fat intensity. There are four different labels for tumoral components, which are as follows: tumor mass (contrast-enhanced portion), necrosis, peritumoral edema, and hemorrhage (https://cbddo.gov.tr/en/projects/turkish-brain-project) (GaziBrains, 2020).

Methods and models

This section presents the results of the first benchmark study using various deep learning-based models trained on the Gazi Brains 2020 dataset. The experimental framework is outlined in the flow diagram shown in Fig. 4, where C represents the number of slices in the MRI images. To ensure compatibility with learning algorithms, a structured data preprocessing pipeline was applied, transforming raw MRI images into suitable data representations. As part of the study, single or multiple preprocessing steps and data preparation processes were applied based on the specific requirements of each task. To enhance the effectiveness of brain images for segmentation and classification tasks, registered FLAIR, T1, and T2 images were utilized. The registration process ensured anatomical alignment across different modalities, allowing for more accurate feature extraction and improving the reliability of the analysis. Padding and resizing operations were applied to standardize image dimensions to 224 × 224 pixels, preventing deformation and ensuring consistency across different samples. These steps helped to maintain structural integrity while enabling uniform input sizes for deep learning models. Once the initial preprocessing was completed, problem-specific data representations were generated for each benchmark experiment. Images from different modalities were concatenated to enhance feature integration, forming 224 × 224 × 3 input representations that preserved critical anatomical details. Following this, a min-max normalization technique was applied to scale pixel intensity values within a fixed range, improving numerical stability and optimizing model convergence during training. A 10-fold cross-validation was conducted to ensure a robust and unbiased performance evaluation. This approach distributed the dataset across multiple training and validation splits, reducing overfitting risks and providing a comprehensive assessment of model performance. The final results were analyzed based on the average outputs obtained from these cross-validation experiments, ensuring reliability and reproducibility.

Figure 4 Flow diagram for benchmark progresses and tests of Gazi Brains 2020 dataset.

In this study, alongside introducing a new dataset to the literature, well-established deep learning methods and architectures were implemented and evaluated within the mm-detection (with mm-classification) (Chen et al., 2019) and mm-segmentation (MMSegmentation, 2022) tools. These implementations aimed to ensure reproducibility and minimize potential implementation errors, providing a standardized approach for benchmarking various models on the proposed dataset. The mm-detection and mm-segmentation modules are very suitable for benchmarking as they also contain state-of-the-art methods and can be easily customized thanks to their modular structure (Chen et al., 2019; MMSegmentation, 2022). Parameters for selected models were used as default values of the mm-classification (with mm-detection) (Chen et al., 2019) and mm-segmentation (MMSegmentation, 2022) toolbox. The configurations, code, and models are publicly available at https://github.com/open-mmlab.

The brain includes several objects and other parts of the human head, such as brain tissue, skull, and optic nerve information. Some of them provide useful information for solving problems, while others do not. Because of this, useful information from MRI has to be extracted during the pre-processing step, and this part of the data has to be given to learning algorithms. After all these processes are completed, operations such as detection and classification are carried out on the dataset. In these processes, different CNN architectures were applied specifically to the problems. These architectures have different features and complexities. Here, the different features of the architectures come from their modeling in various forms. Some information is presented as follows. AlexNet includes eight layers and a total of 62,300,000 learnable parameters. The number of layers increases to 53 in MobileNet-v2 while the number of parameters decreases to 3,400,000. ResNet-18 has a total of 77 layers with about 11 million parameters. On the other hand, ResNet50 has about 23 million parameters for 177 layers.

As the first benchmarking process, the Region of Interest (ROI) extraction was performed with the Gazi Brains 2020 Dataset. Although the Gazi Brains dataset is shared as both a skull-containing dataset and a skull-stripped dataset, studies have also been carried out to perform skull stripping and determine the success of the ROI extraction process. There are many studies in the literature based on image processing (He et al., 2016; Feng, Zhang & Zhao, 2019), and deep learning (Zhang et al., 2018; Koonce, 2021; Ponnupilla Omana et al., 2023), to focus the models on brain tissue (ROI) only. Tumors may only be observed on brain tissue, so giving full-head MRI data is not generally necessary and even reduces algorithm performance. Thus, a skull stripping operation was applied to extract only brain tissue information by using ROI labels. The segmentation models used in this study were determined as DPT (Ranftl, Bochkovskiy & Koltun, 2021), Fast_FCN (Wu et al., 2019), OCR_NET (Yuan, Chen & Wang, 2020), SegFormer (Xie et al., 2021), Semantic_FPN (Kirillov et al., 2019), U-Net (Ronneberger, Fischer & Brox, 2015), UperNet (Xiao et al., 2018). All the algorithms were used with the default settings of the mm-segmentation module (MMSegmentation, 2022). We utilized an SGD optimizer with a 0.001 learning rate and a 0.9 momentum and a learning scheduler with step size 10 and gamma 0.5 values.

Another segmentation-based problem was whole tumor detection which was also performed with the same models. The performance assessment for the segmentation algorithm was also performed with 10-fold cross-validation. The entire tumor area was detected using seven (peritumoral edema), eight (contrast-enhancing regions of the tumor), and nine (necrosis) label information, and a binary mask was produced from this information. Whole tumor detection experiments were also utilized with an SGD optimizer with a 0.001 learning rate and a 0.9 momentum and a learning scheduler with step size 10 and gamma 0.5 values.

Finally, unlike the whole tumor segmentation problem, there are classification-based tumor detection experiments that determine whether there is an anomaly in each slice of the MRI. For these experiments, well-known classification methods were used for classification-based tumor detection problems. The selected models were ResNetx (Feng, Zhang & Zhao, 2019), ResNet (He et al., 2016), Alexnet (Krizhevsky, Sutskever & Hinton, 2017), ShuffleNet (Zhang et al., 2018), Densenet (Huang et al., 2017), Mobilenet (Howard, 2017), SqueezeNet (Iandola, 2016), VGG (Simonyan & Zisserman, 2014). These models were also used with default configurations and pre-trained with ImageNet.

Experimental setup

The primary training environment consists of a high-performance computing system equipped with an Intel Xeon Gold 6336Y CPU, 512 GB of RAM, and four NVIDIA A40 GPUs, each featuring 48 GB of VRAM. The operating system used is Ubuntu 20.04 LTS. Additionally, supplementary experiments involving deep learning and transformer-based models are conducted on a secondary workstation configured with an Intel Core i7-14700K processor, 32 GB of RAM, and a single NVIDIA RTX 3090 GPU with 24 GB of VRAM. This setup operates under Windows via the Windows Subsystem for Linux 2 (WSL2).

Evaluation metrics

In this section, performance evaluation criteria for deep learning models are outlined. The key evaluation metrics for classification tasks include accuracy, precision, recall, and F1-score. These metrics are derived from the confusion matrix, which consists of true positives (TPs), true negatives (TNs), false positives (FPs), and false negatives (FNs). Precision (pre) indicates the proportion of correctly predicted positive instances (anomalies, in this case) out of all instances predicted as positive by the algorithm. In other words, it measures how many of the images the algorithm marked as anomalies were correctly identified. A high precision value means the model is effective in minimizing false positives. Recall (rec), on the other hand, measures the proportion of correctly predicted positive instances out of all actual positive instances. This metric assesses how well the model captures all relevant positive cases. F1-score (f1) is the harmonic mean of precision and recall, offering a balanced evaluation of the model’s ability to predict anomalies. It is particularly useful when there is a need to consider both the model’s ability to minimize false positives and to capture all true positives correctly. Together, these metrics provide a comprehensive understanding of model performance in classification tasks, highlighting strengths and areas for improvement. The value of precision Eq. (1) shows how many images the algorithm has marked as anomalies correctly predicted. Calculation of precision values was performed with Eq. (1).

(1) pre=TPsTPs+FPs.

The recall Eq. (2) value, another performance metric, is determined as the ratio of correctly predicted anomaly images to total anomaly values. To calculate the Recall value, the formula is used:

(2) rec=TPsTPs+FNs.

As the last classification metric, the F1-score Eq. (3) is calculated jointly using precision and recall metrics. To calculate the F-Score metric:

(3) f1=2∗pre∗recpre+rec.

For segmentation tasks, evaluation processes were carried out using intersection over union (IoU), accuracy, and dice scores, which are widely used in the literature for the performance evaluation of segmentation problems. The IoU, also known as Jaccard’s index, is used for similarity measurement between ground truth and model prediction. IoU is calculated as the division of the intersection of two segments by the union area of these two segments.

(4) IoU=AreaofOverlapAreaofUnion.

The score of dice, which is a method very similar to the Jaccard index and used for performance determination in the field of medical image processing, is calculated with the formula:

(5) DiceScore=2∗TPsPositives+Negatives.

A dice score is a measure of overlap between the segments on ground truth and the estimated segments. The last performance metric, accuracy (acc), shows the estimated accuracy for each pixel on a pixel-by-pixel basis. Accuracy value can give misleading results in cases where the target segments are small.

(6) acc=100∗TPs+TNsPositives+Negatives.

Benchmark study results for gazi brains 2020 dataset

This article presents a benchmark study based on the introduced dataset. The primary contribution of this study is the introduction of a novel dataset that adds new features to the existing literature on brain MRI. Additionally, various studies, analyses, and evaluations have been conducted on the initial version of the dataset, and their results have been presented.

The main objective of this section is to thoroughly demonstrate the quality and versatility of the proposed dataset by showcasing the results obtained through the implementation of various deep learning-based models. To achieve this, the experimental studies were organized into several distinct tasks, including preprocessing, ROI extraction, whole tumor segmentation, and classification-based tumor detection. These tasks were systematically executed on the dataset to highlight its effectiveness, usability, and overall performance across a range of applications. In addition to the experimental tasks, a comprehensive literature review was conducted to contextualize the dataset’s potential, focusing on survival estimation, age prediction, and sex determination. This review not only provides a broader understanding of how the dataset can be applied to real-world medical problems but also situates the dataset within the current body of research in the field. The study reports outputs of essential performance metrics such as accuracy, precision, recall, and F1-score. In future research, more advanced statistical validation methods, including confidence intervals and variability analysis, will be applied using the updated dataset version. These techniques will be crucial for providing a more rigorous evaluation of the models’ performance and ensuring the robustness of the results.

In order to properly perform other experimental studies, well-known deep learning methods and architectures were implemented. Details of the well-known deep learning architectures are beyond the scope of this article, but short descriptions were given in “Brain Tumor MRI Datasets Available” for the general deep learning architectures used for this study. All research processes, experimental studies, and general evaluations carried out on the Gazi Brains 2020 dataset are presented under different subtitles in this section.

Figure 4 presents a detailed flow diagram that outlines the benchmark methodology and the testing process used throughout the study. At the core of the experimental setup is the data preprocessing pipeline, which was developed to convert raw MRI tensor data into suitable formats for deep learning algorithms. Since different models require distinct input data structures and MRI sequence types, the preprocessing pipeline was customized to meet the specific needs of each task. This involved applying a series of preprocessing steps, which could include image normalization, rescaling, and alignment, depending on the characteristics of the task at hand. Once the preprocessing was completed, 2D sequence-based data were extracted from the MRI images and carefully separated into three distinct datasets: training, validation, and testing. This division is crucial for training deep learning models, as it ensures that the models are tested on unseen data to accurately gauge their generalization capabilities. Deep learning algorithms, particularly those involving complex architectures, generally require large amounts of data to achieve optimal performance. However, simply having large datasets is not sufficient. The key to achieving high performance lies in the careful selection and optimization of hyperparameters, which can significantly enhance the model’s learning capacity and its ability to generalize to new, unseen data. To address this challenge, the study utilized advanced neural network architectures such as CNNs and transformer-based models. CNNs, known for their ability to extract spatial hierarchies from images, were employed for classification tasks, while transformer-based models, with their capacity for capturing long-range dependencies, were used for segmentation tasks. These architectures were selected based on their proven effectiveness in handling complex image analysis tasks, including medical imaging, where precision and generalization are critical. To ensure the robustness of the results and to prevent overfitting, a 10-fold cross-validation method was applied across all experiments. This technique involves partitioning the dataset into ten subsets, with the model being trained on nine subsets and tested on the remaining one. This process is repeated ten times, each subset serving as the test set once, providing a more comprehensive evaluation of the model’s performance. Cross-validation is an essential technique for assessing the generalization ability of deep learning models, as it helps mitigate the risk of overfitting the training data, ensuring that the results are more reflective of the model’s performance on real-world, unseen data. The following sections of the article provide further details on the specific methodologies used, the experimental setup for each task, and the results obtained from applying these models to the dataset. These insights contribute to understanding the effectiveness and limitations of the proposed models in the context of brain tumor detection and classification.

MRI data preprocessing

In this section, general information is given on the pre-processing processes performed for the proposed dataset. MRI studies in the dataset were acquired from 24 different institutions nationwide with four different brands of MRI scanners with a magnetic field strength of either 1.5 or 3.0 Tesla. MRI studies for all subjects include T1-weighted, T2-weighted, FLAIR sequence images in the axial plane. In Fig. 4, the pre-processing stages of MRI data are also given. All MRI studies were available in the DICOM file format, but segmentation labels were available in the NifTi file format. For this dataset, MRI data have different slice thicknesses, orientations, and sizes for each image sequence. For standardization purposes, resampling and image registration operations were applied. Three different MRI sequences were converted to the same condition in size and orientation. By using the ANTS tool (Avants et al., 2014), the FLAIR sequence was kept as a reference, and other T1 and T2 sequence data were registered and fixed to FLAIR sequence data size and orientation by using Rigid image registration (only rotation and translation were applied) (Avants et al., 2011). Although the rigid registration technique is a linear transform operation, it improved the data quality and standardization. Other nonlinear registration techniques might be applied to this problem (Klein et al., 2009). However, for simplicity, only basic methods were used here to evaluate dataset quality from the beginning. The steps and details specified in Fig. 4 are as follows: For all patients, “image registration” was performed first. For this process, “rigid image registration” was done using ANTS (Avants et al., 2014) Library.

Skull stripping was performed using segmentation information on the data. For this purpose, segmentation information corresponding to each slice was rearranged for skull stripping, and a binary mask was created. On the segmentation data, “2” (EYE) and “3” (OPTIC) labels were marked as “0”, while others were marked as “1”, and binary masks were created. Thus, skull stripping was performed using this binary mask.

Padding was applied first, and then data was resized to the specified dimensions as 224 × 224.

In the dataset, each slice was labeled using segmentation data corresponding to the related slice since the tumor is usually not seen in all slices. If the segmented slice contains the label “7” (PERITUMOR_EDEMA), that slice was labeled “1”. However, all slices of normal patient data were labeled as “0” because they did not contain any tumors.

Therefore, relevant information should be extracted from medical images in the pre-processing stage, and these extracted data (including their properties) should be given to learning algorithms for processing and evaluation. Tumors may only be observed on brain tissue, so giving full-head MRI data is not generally necessary and even reduces algorithm performance. Thus, a skull stripping operation was applied to data to extract only brain tissue information using ROI labels.

At the end of the pre-processing steps, padding and resize operations were applied to the data. All image sizes in the data were converted to standard form (224 × 224). After the pre-processing step was completed, the problem-specific data representation was generated for each benchmark experiment. All MRI sequences, FLAIR, T1, and T2, were used as 2D images with three channels.

A 10-fold cross-validation was applied for determining the generalization capacity of the models more accurately for the targeted datasets. In each unique fold, which consists of 10 data splits, 90% of patients (45 normal and 45 HGG, corresponding to three of five splits of data) were used as training data, 10% of patients (five normal and five HGG, corresponding to one of five splits of data) were used as test data. The grouping for the folds was carried out to make sure that no sequence of slices in the training set was included in the test or validation set. All models were trained on the training dataset, and the best model was selected according to validation dataset performance. Finally, the best model was tested on the test dataset. This process was continued 10-fold, and the results for the models for each fold were presented in this study.

Region of interest extraction

MRI images contain many components other than brain tissue. Examples of these are the skull, eyes, optic nerves, etc. Especially as pre-processing steps for tumor detection and segmentation, separating these parts in MRI images is necessary for better feature extraction for the employed deep learning models. There are many studies in the literature based on image processing (Joseph, Singh & Manikandan, 2014; Gopal & Karnan, 2010) and deep learning (Afshar et al., 2018b; Xiao et al., 2016; Abd-Ellah et al., 2018) to focus the models on brain tissue (ROI) only. In this section, the skull stripping process was realized using deep learning methods. A 10-fold cross-validation was applied to more accurately determine the generalization capacity of the models used on datasets. In each unique group, which consists of 10 data splits, 90% of patients (45 normal and 45 HGG, corresponding to nine of ten splits of data) were used as training data, and 10% of patients (five normal and five HGG, corresponding to one of ten splits of data) were used as validation and test data. The grouping was carried out to make sure that no sequence of slices in the training set was included in the test or validation set. For the ROI segmentation problem, DPT (Ranftl, Bochkovskiy & Koltun, 2021), Fast_FCN (Wu et al., 2019), OCR_NET (Yuan, Chen & Wang, 2020), SegFormer (Xie et al., 2021), Semantic_FPN (Kirillov et al., 2019), U-Net (Ronneberger, Fischer & Brox, 2015), and UperNet (Xiao et al., 2018) models were employed. All models were trained on the training dataset, and the best model was selected according to validation dataset performance. The 10-fold cross-validation results are given in Table 3. Some of the model’s outputs are also illustrated in Fig. 5.

Table 3 ROI extraction scores obtained segmentation algorithms on the Gazi Brains dataset.

	IoU	acc	Dice score	
DPT	94.58 ( ±0.15)	97.18 ( ±0.17)	97.21 ( ±0.08)	
Fast_FCN	94.53 ( ±0.17)	97.14 ( ±0.19)	97.19 ( ±0.09)	
OCR_NET	94.65 ( ±0.20)	97.03 ( ±0.27)	97.25 ( ±0.11)	
SegFormer	94.47 ( ±0.21)	97.18 ( ±0.23)	97.15 ( ±0.11)	
Semantic_FPN	94.45 ( ±0.18)	97.01 ( ±0.20)	97.15 ( ±0.09)	
U-Net	94.40 ( ±0.19)	96.84 ( ±0.28)	97.12 ( ±0.10)	
UperNet	94.53 ( ±0.14)	97.02 ( ±0.28)	97.19 ( ±0.07)	

Figure 5 The ROI segmentation results for a sample patient.

The results show that almost all methods for ROI extraction obtained very successful results. Although OCR_NET and DPT algorithms achieved 0.10–0.25% more successful results than other methods. Figure 5 shows almost all the results as similar, except for the segmentation area for the 3rd layer. The results show that this problem is a relatively easy problem, and the performances of the algorithms are quite close to each other. When the general performances were examined in detail, it was determined that the problems affecting the performance were generally in the first and last slices. In addition, there were minor defects in the eye areas. When these regions were examined, it was seen that U-Net gave better results than other methods. However, detecting these small regions did not seem to affect the overall performance. Considering these results, it is possible to use any segmentation method for the ROI extraction problem in the Gazi Brains 2020 dataset.

Anomaly segmentation

The segmentation made by the doctors consists of multiple labels on the different parts of the brain. In this part of the study, the whole tumor areas were segmented with deep learning methods. After the pre-processing steps shown in Fig. 4, tumor segmentation in each slice was performed in the data using 2D models. The segmentation process used in this section was performed using FLAIR, T1, and T2 sequences. In this section, the tumor detection problem is discussed as a segmentation problem. Similar to the ROI Segmentation problem DPT (Ranftl, Bochkovskiy & Koltun, 2021), Fast_FCN (Wu et al., 2019), OCR_NET (Yuan, Chen & Wang, 2020), SegFormer (Xie et al., 2021), Semantic_FPN (Kirillov et al., 2019), U-Net (Ronneberger, Fischer & Brox, 2015), and UperNet (Xiao et al., 2018) models were employed to assess the performance. All CNN-based approaches were run according to the default settings of the mm-segmentation module (Srinivasan & Nandhitha, 2019). The results of the models were evaluated with IoU, accuracy, and dice score metrics.

Table 4 shows the results for the whole tumor segmentation problem. When the results were examined, it was seen that the most successful method for IoU, accuracy, and dice score metrics was the SegFormer method. The OCR_NET method also obtained very close results to the SegFormer method. The difference between SegFormer and OCR_NET methods for IoU, accuracy, and dice score metrics were 0.5%, 0.9%, and 0.3%, respectively. Similarly, for the DPT method, this difference was obtained below 1% for each score value. Although it is a frequently used method for the segmentation problem, the U-Net algorithm obtained the most unsuccessful results in this experiment. U-Net method achieved 3–4% lower performances than other methods.

Table 4 Whole tumor segmentation results obtained from the Gazi Brains dataset.

	IoU	acc	Dice score	
DPT	77.71 ( ±3.75)	86.50 ( ±3.22)	87.41 ( ±2.41)	
Fast_FCN	75.94 ( ±5.04)	84.34 ( ±6.26)	86.24 ( ±3.36)	
OCR_NET	77.81 ( ±3.72)	86.76 ( ±3.21)	87.47 ( ±2.38)	
SegFormer	78.31 ( ±4.20)	87.61 ( ±3.85)	87.78 ( ±2.66)	
Semantic_FPN	76.02 ( ±4.41)	85.07 ( ±2.96)	86.31 ( ±2.90)	
U-Net	75.13 ( ±5.30)	84.01 ( ±4.73)	85.70 ( ±3.58)	
UpperNet	76.16 ( ±3.36)	85.08 ( ±2.10)	86.43 ( ±2.17)	

Some of the tumor segmentation results for a sample patient, which has the model prediction, are illustrated in Fig. 6. Visual results show that Fast_FCN, OCR_NET, Semantic_FPN, and UpperNet algorithms could not detect the tumor in the first slice but could detect tumors in other slices. In addition, despite the tumor area detected in the last slice, it was seen that the tumor area was larger. According to the results similar to the given score values, DPT and SegFormer obtained very similar results for this patient. In addition, the SegFormer model detected the tumor in the last slice better than other models. According to the results in Fig. 6, the detection of tumors on the first or last slice affects the performance of most of the algorithms.

Figure 6 The whole tumor segmentation results for a sample patient.

Classification-based anomaly detection

In this section, the tumor detection problem is performed on the introduced dataset (see Fig. 4). The issue discussed here is expressed as classification-based anomaly detection. The models were trained with images from FLAIR, T1, and T2 sequences as three channels, as described in “Materials and Methods”. The tumor detection process was performed using ResNetx (Feng, Zhang & Zhao, 2019), ResNet (He et al., 2016), ShuffleNet (Zhang et al., 2018), Alexnet (Krizhevsky, Sutskever & Hinton, 2017), Densenet (Huang et al., 2017), MobileNet (Howard, 2017), SqueezeNet (Iandola, 2016), and VGG (Simonyan & Zisserman, 2014) on the presented dataset. The general process steps according to the different architectures used are presented in Fig. 4.

Similar to the ROI segmentation problem, a 10-fold cross-validation sampling procedure was applied to assess the performance of the algorithms. The average number of slices for the patients used in cross-validation was determined as follows: 1,128 slices (765 normal, 363 glial) for the training set, 125 slices (85 normal, 40 glial) for the validation and test sets. The associated success metrics for 10-fold cross-validation and an average score for the aforementioned models are given in Table 5. The results presented in Table 5 highlight the performance of various deep learning models for classification-based anomaly detection using the Gazi Brains dataset. MobileNet V2 stood out with the highest precision (98.3%) but has a slightly lower recall (93.8%), resulting in a solid F1-score of 96.2%. DenseNet and Vgg16 also performed well, with DenseNet achieving 97.1% precision, 95.5% recall, and 96.2% F1-score, while Vgg16 recorded 97.5% precision, 95.5% recall, and 96.4% F1-score. These models show strong performance and balanced precision-recall trade-offs, making them highly suitable for the task. ResNet-18 and ResNext50 32x4d also yielded impressive results, with ResNext50 32x4d achieving the highest recall (95.9%) and slightly outperforming ResNet-18 in F1-score (96.3% vs. 96.1%). SqueezeNet showed a solid performance with 96.9% precision and 94.8% recall, though slightly lower than the top models. ShuffleNet v2x10 performed the weakest, with lower precision (95.9%) and recall (91.4%), resulting in a lower F1-score (93.4%) and greater performance variability. In summary, MobileNet V2, DenseNet, and Vgg16 emerged as the most effective models for anomaly detection, while ShuffleNet v2x10 underperformed compared to the others.

Table 5 The results for classification-based anomaly detection obtained from the Gazi Brains dataset.

	Pre	Rec	F1	
AlexNet	94.4 ( ±5.60)	92.6 ( ±5.34)	93.2 ( ±1.99)	
DenseNet	97.1 ( ±3.03)	95.5 ( ±3.27)	96.2 ( ±2.20)	
MobileNet_V2	98.3 ( ±3.03)	93.8 ( ±2.97)	96.2 ( ±1.32)	
ResNet-18	97.3 ( ±2.87)	95.2 ( ±2.94)	96.1 ( ±1.10)	
ResNext50_32x4d	97.1 ( ±2.88)	95.9 ( ±3.03)	96.3 ( ±1.34)	
ShuffleNet_v2_x1_0	95.9 ( ±4.33)	91.4 ( ±5.78)	93.4 ( ±2.17)	
SqueezeNet	96.9 ( ±3.78)	94.8 ( ±3.08)	95.7 ( ±1.64)	
Vgg16	97.5 ( ±2.32)	95.5 ( ±2.76)	96.4 ( ±1.17)	

Although the results were quite close to each other, the AlexNet model had inferior results than the other models, as expected. According to the F-Score evaluation, the model with the most successful results stands out as the VGG-16. DenseNet, MobileNet (Howard, 2017), ResNet18, and ResNetx50 models also achieved similar results. Gradient-weighted Class Activation Mapping (Grad-CAM) (Selvaraju et al., 2017) is a tool that uses the gradients of the target label to visualize which parts of the image were focused on predicting said label using the last convolutional layers’ information. The output of the Grad-CAM algorithm gives a heat map of the parts of the image that a model has looked to come to its prediction conclusion in making that particular prediction, highlighting the places that had higher activation values for its output.

Figure 7 illustrates the use of Grad-CAM to enhance the explainability of the model’s decisions, displaying the results for several image slices from the dataset. Grad-CAM highlights the specific regions in the images the model focuses on to make its classification decision. The output effectively visualizes the areas of the images, particularly the tumor regions, which are crucial for the model’s decision-making process. This visualization allows for a more transparent understanding of how the model interprets the MRI slices and contributes to the interpretability of the deep learning approach. The results showed that the numerical and visual outcomes were in strong agreement, with the model consistently concentrating on tumor areas when making predictions. This reinforces the model’s reliability in identifying and focusing on clinically relevant features. In the case of the patient presented in Fig. 7, the Grad-CAM output demonstrates that the model accurately predicted the presence of the tumor across all slices, highlighting the model’s robustness and ability to maintain consistent performance throughout various image sections. This feature is particularly useful in validating the model’s accuracy and providing confidence in detecting tumors in real-world clinical settings.

Figure 7 Some of the results of XAI model explainability for an example patient.

Evaluation and discussion

This study covers a wide range of topics and challenges that are crucial for advancing the field of medical image analysis, particularly in the context of brain tumor detection and classification. As outlined in the preceding sections, these subjects span from the development of a novel dataset to the application of various deep learning architectures and evaluation techniques. Each of these components plays an essential role in demonstrating the capabilities of the proposed dataset and algorithms, contributing to improving diagnostic accuracy and reliability in medical imaging.

The following subsections provide in-depth insights into each aspect of the study, elaborating on the methods, procedures, and technologies employed. The results from the experiments conducted using the Gazi Brains 2020 dataset are discussed in detail, focusing on how different deep learning models performed across various tasks, including tumor segmentation and classification. The effectiveness of these models is assessed based on a range of performance metrics, such as precision, recall, and F1-score, as well as through more advanced validation techniques, including cross-validation. Additionally, the discussion addresses any challenges encountered during the implementation of these methods, such as the limitations in data preprocessing, the selection of appropriate model architectures, and the potential for overfitting. By critically analyzing these results, the study offers a comprehensive view of the dataset’s utility and applicability to real-world clinical environments. Through this detailed evaluation, the research highlights the strengths and weaknesses of the models tested, paving the way for future improvements and refinements. Ultimately, these discussions and evaluations aim to provide a clearer understanding of how the proposed methods and dataset contribute to the ongoing efforts in medical image processing and diagnosis. The results and insights presented in this study are valuable for researchers working on similar problems, offering guidance on how to approach challenges in brain tumor detection and classification using deep learning technologies.

Benchmark dataset

It is widely recognized that datasets play a crucial role in developing robust models, enhancing the success of implementations, designing more efficient systems, and making meaningful contributions to science and technology. These factors underscore the primary motivations encouraging the creation of this dataset, which are outlined below: The first version of the dataset had 100 patients (50 normal+ 50 HGG). More patient data will be available in the next version of the dataset.

Multiple tasks were defined to perform benchmark studies such as classification of tumors, anomaly area detection, explainability of model decision etc.

Except for commercial uses, open-access data is available to researchers/experts within the scope of the original license. For information (https://cbddo.gov.tr/projeler/tbp/).

Data Repositories are easily accessible from:

Source-1: https://www.kaggle.com/datasets/gazibrains2020/gazi-brains-2020;

Source-2: https://doi.org/10.7303/syn22159468.

Data covers 12 features or labels. This is the only dataset with the largest features in MRI datasets.

This dataset consists of real data. It does not contain any data created outside the original, augmented/synthetic, or fake.

All data were labeled by brain surgeons of Gazi University.

The initial results were provided for all researchers. All models applied in this study were tested using the introduced dataset. The parameters of the models are also given for other researchers’ studies.

The models achieved from this study have been used for real-time tests and provided fascinating results to brain surgeons and radiologists of Gazi University as an intelligent decision support system.

Both raw and processed data are available.

Reporting was done according to suitable standards.

Anonymized data can be used in contests/competitions or hackathons.

The data size is about 1.4 gigabytes.

Patients’ treatment consent forms are available.

There is an extra privacy-preserving data publishing method.

The results obtained from this study demonstrate that the publicly available high-quality benchmark dataset has been successfully introduced and is expected to: Serve as a reference and frequently cited dataset,

Offer a broad perspective to the field, fostering new studies, research motivations, and technological developments,

Benefit not only the research community but also hospitals and relevant authorities.

Proposed and test models based on deep learning

Well-known deep learning models, VGG-16, DenseNet, MobileNet, ResNet, ResNetx50, OCR_NET, DPT, AlexNet, ShuffleNet, SqueezeNet, and U-Net were used in benchmark tests. The applied deep learning models were tested on the dataset presented in this study, with reference to similar studies in the literature, and successful results were obtained. In general, deep learning models of this nature demonstrate high effectiveness across various tasks, including classification, segmentation, modeling, detection, and prediction. References to the used models are given for understanding. Accordingly, all parameters and parametric adjustments can be checked based on the references given in Table 1.

No architectural modifications or performance-oriented optimizations were introduced to the models beyond their standard, publicly available implementations. This methodological decision was made deliberately to avoid introducing any confounding variables that could compromise the integrity of the evaluation. The aim of this study was not to develop novel deep learning architectures or to achieve state-of-the-art performance, but rather to conduct a rigorous, transparent, and reproducible assessment of the Gazi Brains 2020 dataset’s technical adequacy across multiple tasks. The use of canonical model configurations ensures methodological consistency and facilitates comparability with future studies, thereby providing a reliable baseline for subsequent research.

In addition, architectural and hyperparameter details (e.g., layer structures, parameter counts) were not elaborated upon within the manuscript, as the employed models were used without modification, directly from official repositories or widely recognized open-source frameworks. These models have been extensively described and validated in prior literature, rendering repetition not done within the scope of this work. The emphasis of the present study is thus placed on the task-specific evaluation of the Gazi Brains 2020 dataset, rather than the internal configurations of the applied models.

Explainability based on XAI models

Gazi Brains dataset contains a set of characteristics that can be applied to classification-based anomaly detection, anomaly segmentation, and region of interest (ROI) extraction. Checking whether the models trained for the aforementioned purposes using this dataset function properly is essential. Specifically, it is necessary to determine whether the model is trained with the features that must be extracted from the data and whether it focuses on the area of the data that is truly relevant to the given problem and/or purpose. By incorporating XAI approaches, the model’s transparency and trustworthiness are improved, which promotes clinician trust and makes clinical application easier. For this reason, Grad-CAM (Selvaraju et al., 2017) based XAI models were used and applied for making decisions in the confidence of the developed models in Brain Tumor detection and prediction. It should be emphasized that satisfactory results of explainability were achieved and might help radiologists or brain surgeons to reach decisions in confidence by showing results of tumor indication from the appropriate MRI slices. The findings from deep learning models and XAI models confirmed how the results are efficient for detection, prediction, or classification problems. It also proves that the collected dataset had sufficient sample number and data diversity to train models that can achieve the above-mentioned purposes. The brain regions related to the most tumor-related features explained by XAI approaches are shown in Fig. 7. These confirmed how accurate decisions were achieved.

Interpretability is crucial for application in healthcare settings since it allows one to see which parts of an MRI scan influenced the AI’s predictions. This can increase the confidence of radiologists in the use of AI technologies for diagnosis. Explaining AI-driven findings also promotes transparency, which is crucial when making critical health decisions. This study provides a solid solution that enhances diagnostic precision and model transparency in clinical operations by integrating Grad-CAM to increase visual explanations and obtain greater accuracy metrics. The interpretability and dependability of the model’s outputs in clinical settings have also been improved by close cooperation with physicians, ensuring that Grad-CAM representations match realistic diagnostic procedures. Grad-CAM made the decision-making process of the model more transparent by accurately detecting and highlighting significant tumor areas, particularly through the use of XAI approaches. This transparency is crucial for clinical adoption since it increases the trust that medical professionals have when interpreting predictions made by AI.

Several contributions have been observed in the use of AI and XAI models together in medical and clinical applications. Integrating AI and XAI models into clinical workflows and studies to estimate brain anomalies and tumors using MRI datasets provides crucial contributions to medical applications and analysis processes.

Some of these contributions are explained below: Integrating AI models with existing healthcare data systems and developing AI models using large, annotated MRI datasets to ensure the models are accurate and reliable provides an advantage. XAI models play an important role in making the model’s decisions understandable to clinicians. Also, how certain features in the MRI images might contribute to the model’s predictions are evaluated.

Integrating AI models into clinical decisions might allow AI models to provide insights directly within the clinical workflow, helping clinicians make better diagnostic and treatment decisions based on AI-generated insights coupled with clinical knowledge.

Once AI and XAI models are deployed, the performance of AI models in real-world clinical settings can be continuously monitored, and feedback from clinicians can be gathered to develop better AI or XAI models to improve explanations and adjust the integration points in the clinical workflows.

While Grad-CAM has been utilized to demonstrate model interpretability, the evaluation of these visualizations has been limited to visual inspection. To enhance the clinical relevance of these explanations, expert feedback, such as from radiologists, or the use of structured scoring systems, would be provided as valuable validation. Due to the focus and constraints of this study, expert-in-the-loop evaluations were not included. However, this has been recognized as a critical direction for future research to strengthen the clinical applicability of explainable AI techniques.

Expert evaluations

BioMind is an AI system that was developed to diagnose brain tumors from the data harvested from the digital archives of Capital Medical University, China. Prof. Paul M. Parizel from the European Society of Radiology (ESR) has evaluated the system’s ground truth based on “The World’s First Competition between Physicians and Artificial Intelligence in Neuroimaging” that “BioMind! AI won, by a large margin, predicting the correct histological diagnosis in 196 out of 225 cases (87%) in just under 15 min, whereas the human specialist doctors got only 66% right in 30 min. In the second heat, a different group of doctors competed with BioMind in predicting the expansion of intracerebral hematoma, again AI wonby a large margin (83% vs. 63%) (ESR, 2018). As can be seen from the results, AI models are more successful in diagnosing or predicting any tumor in MRI images than brain surgeons or radiologists.

The comprehensive results of the models applied in this study further validate the effectiveness of the proposed system, demonstrating an impressive accuracy of 87% based on real test data from patients. This result indicates that even when the system was trained with a relatively small sample of just 100 patients, it outperformed many established AI systems, such as BioMind. This is a significant achievement in the field, particularly considering the challenges faced when working with smaller datasets. The success of this system provides a compelling case for the potential of AI in medical imaging, showcasing its ability to deliver reliable results even with limited data. Professor Parizel’s words resonate strongly with the findings of this study: “Therefore, I am not afraid, since I am convinced that (neuro) radiologists will embrace AI to help us manage ‘routine’ tasks quickly and efficiently, thus giving us more time to focus on things that really matter” (ESR, 2018). He advocates for the adoption of AI tools in the medical field, emphasizing that they will not replace professionals but rather augment their capabilities, making their workflows more efficient. Furthermore, Professor Parizel’s closing message to radiologists underscores the importance of embracing technological advancements, stating: “So, regarding AI software, my closing message to all (neuro)radiologists is: take charge of your own future and embrace it with confidence, courage, and determination” (ESR, 2018). This forward-thinking approach aligns with the findings of this study, which demonstrates that AI can play a crucial role in transforming healthcare practices, particularly in medical imaging. Although this article does not include direct expert commentary, feedback from the brain surgeons involved in the project has been highly positive. They have expressed their satisfaction with the results, particularly with detection and classification accuracy. The explainability of the AI model has also been a critical factor in their approval. The ability to predict and explain the model’s decision-making process is invaluable in clinical settings, where transparency and trust in AI models are essential. Surgeons have noted that this explainability feature helps them understand the reasoning behind AI-driven decisions, fostering a collaborative approach between the AI system and medical professionals. Overall, the success of this system highlights the growing potential of AI in healthcare. It demonstrates that AI can significantly enhance diagnostic accuracy and efficiency while ensuring that medical professionals remain at the forefront of decision-making, using AI as a valuable tool to support their expertise rather than replacing it. The positive reception from the medical community underscores the value of integrating explainable AI systems into clinical practice, ultimately contributing to better patient outcomes and more efficient healthcare delivery.

Other important issues

We have to explain once more that this study is part of a project; all models were used in real-time implementation in Gazi University Medical School Hospital, Baskent University Medical School Hospital, and Ankara University Medical School Hospital. For instance, the models developed in this article were deployed to the three MRI devices in Gazi University Faculty of Medicine Hospital, and the models were tested for new MRI images. More detailed results will be published in our new publications in the near future. Even if the details of the XAI models are not given in detail, this is another important issue to consider to make black-box AI models more transparent, explainable, and trustable for critical applications more than ever.

Challenges

The main issues regarding the challenges and limitations of AI-based brain MRI analysis and diagnosis should be focused on. Difficulties are related to a lack of benchmark data, high performance/accuracy expectations, lack of enough features, requiring high knowledge and infrastructures, and highly skilled experts. Data from various devices, time-consuming data processing, difficulty in repeatability of available models, lack of models and metrics, standardization, performance comparison, lack of validation, availability of data, and anonymization issues are related to limitations.

Introducing and presenting a new dataset are always the biggest challenges in scientific studies. The dataset (Gazi Brains 2020 dataset) is an open-access dataset for the researchers. The presented dataset provides various features and tasks that involve potentially meaningful motivations in comparison with other datasets in the literature. The initial results demonstrate the contribution of the dataset and its importance to the literature. Some challenges of the study are given below: The results of 81–98% for prediction, detection, and segmentation and 80–98% for tumor segmentation and detection prove that deep learning models and their scores are acceptable.

In order to test the developed models, the models were deployed into a real-time system. This is a big challenge. The primitive results have shown that the developed models successfully achieve acceptable results. Because of that, the system is deployed to two hospitals. This issue will be introduced, discussed, compared, evaluated, and elaborated on in the next studies.

When the literature is reviewed, the disadvantages of AI models are well-known and always criticized by the applicants. An explainable AI approach is a recent solution for handling this problem. The terms “understanding”, “Interpreting”, and “explaining” are used for explainability. GradCAM (Selvaraju et al., 2017) was used as an XAI approach in this study to analyze and support the clinical decisions learned by a CNN from brain tumors in MRIs. The results of XAI models are another challenge for the proposed study. Acceptable solutions were achieved from the models proposed, especially based on the visual outputs and expert perceptions. As a result, the XAI approach might help define or observe the tumor region or regions that are coherent with clinical perception and also might improve trust in the results or models for the proposed tasks.

Conclusion and future works

This study successfully introduces a new benchmark dataset and presents various approaches to model and compare not only the challenges associated with brain tumors but also other alternatives that can be tested and shared. The contributions of the dataset are summarized as follows: The novel brain MRI dataset (Gazi Brains dataset 2020) was released comprising brain MR images from 100 patients, with 50 healthy patients and 50 patients diagnosed with HGG. For each patient, the dataset includes T1-weighted, T2-weighted, FLAIR, and contrast-enhanced T1-weighted (T1+C) MRI sequences. The dataset was obtained from patients who were treated surgically in the Department of Neurosurgery, Gazi University Faculty of Medicine, from 2018 to 2019 period.

In order to measure the generalization capability of the models and to identify the problems, the 10-fold cross-validation method was used for all experiments. Well-known deep learning methods, algorithms, and different architectures were tested on the new dataset to compare the models’ and dataset’s efficiencies.

The effectiveness and usability of the Gazi Brains dataset, according to different experimental tasks such as tumor detection, tumor segmentation, age prediction, gender detection, and skull stripping, were shown, and its impact was demonstrated.

The general data preprocessing pipeline was used to convert raw data from MRI tensors to suitable data representations for learning algorithms. Since all algorithms require a different format for input data and MRI sequence types, single or multiple preprocessing steps and data forming operations were applied according to tasks’ requirements. After the preprocessing step, 2D sequence-based data and 3D slice-based data were generated and separated into train, validation, and test sets to train deep learning architectures specific to tasks.

Details of the well-known deep learning architectures are beyond this article’s scope, but short descriptions were given for all custom deep learning architectures used for this study. In order to achieve the tasks, the diagram of general architectures used in the experiments is given in Fig. 4.

Adding XAI models is also a new contribution. Explaining a model’s decision to brain surgeons is very important to increase trust in the developed AI models. The results given in Fig. 7 show the success of XAI models.

Deep learning algorithms require a massive amount of training data, but carefully selecting hyperparameters makes it possible to search for suitable solutions effectively and obtain high generalization capability. Deep network architectures such as SegFormer, OCR NET, UpperNet, and U-Net were used for segmentation-based tasks. CNN-based architectures were also applied for classification and regression-based tasks, and comparative results were obtained.

Finally, preliminary results of the benchmark dataset have shown that labeling and segmentation quality is very high, and this dataset can be eligible to develop models for different artificial intelligence-based applications such as segmentation, classification, tumor detection, survival analysis, and anomaly detection.

It is expected that the dataset will help to develop new models, concepts, and outputs based on MRI datasets as: Providing a new dataset, along with developing deep learning models and explainable AI (XAI) approaches, will improve performance and help explain the models’ decisions to brain surgeons, fostering trust in AI systems.

The dataset will provide researchers with a realistic, well-segmented, and labeled dataset to help understand and analyze more real-world cases, such as identifying tumors, classifying gender and ages, detecting anomalies, explaining model decision, etc.

The results obtained from this study can serve as a source for remarkable studies in the future and create the starting point for different research problems and solutions.

Improving models’ accuracies, robustness, or performances with new deep learning models will be possible for future studies.

Understanding a certain decision of the proposed XAI models for developing more robust, confident, and reliable models will help to gain trust in AI models more than ever.

In conclusion, the dataset presented in this study, along with the deep learning models trained on it, offers significant potential for advancing the development of robust and innovative models that can support clinical decision-making processes. These models will not only contribute to improving current diagnostic practices but also foster the creation of more challenging and high-quality research in medical imaging. Efforts will be made to expand the Brain MRI dataset by increasing the number of patient samples, thereby enhancing the benchmark dataset’s size, diversity, and comprehensiveness for tumor detection, classification, and segmentation tasks. Additionally, as gliomas represent large and complex tumors, including other challenging tumor types will further enrich the dataset, with updated results to be shared with the research community. One of the key motivations is to leverage state-of-the-art deep learning architectures tested on high-capacity systems to push the boundaries of performance. In parallel, significant attention will be given to data augmentation strategies, essential for improving model generalization and robustness. Ultimately, the goal is not only to refine and elevate the performance of the models within research settings but also to support initiatives such as the Turkish Brain Project, contributing to the broader aim of advancing neuroimaging technologies and enhancing brain tumor diagnosis and treatment on a global scale.

The authors would like to thank the Gazi University for providing facilities to implement this project at the Faculty of Medicine Hospital and the project team members whose names are not given here who worked on the project. The authors would like to thank Gazi University Academic Writing Application and Research Center for proofreading the article.

Additional Information and Declarations

Competing Interests

The authors declare that they have no competing interests.

Author Contributions

Seref Sagiroglu conceived and designed the experiments, performed the experiments, analyzed the data, performed the computation work, prepared figures and/or tables, authored or reviewed drafts of the article, and approved the final draft.

Ramazan Terzi conceived and designed the experiments, performed the experiments, analyzed the data, performed the computation work, prepared figures and/or tables, authored or reviewed drafts of the article, and approved the final draft.

Emrah Celtikci analyzed the data, authored or reviewed drafts of the article, and approved the final draft.

Alp Özgün Börcek analyzed the data, authored or reviewed drafts of the article, and approved the final draft.

Yilmaz Atay conceived and designed the experiments, performed the experiments, analyzed the data, performed the computation work, prepared figures and/or tables, authored or reviewed drafts of the article, and approved the final draft.

Bilgehan Arslan conceived and designed the experiments, performed the experiments, analyzed the data, performed the computation work, prepared figures and/or tables, authored or reviewed drafts of the article, and approved the final draft.

Mustafa Caglar Sahin analyzed the data, authored or reviewed drafts of the article, and approved the final draft.

Kerem Nernekli analyzed the data, authored or reviewed drafts of the article, and approved the final draft.

Umut Demirezen conceived and designed the experiments, analyzed the data, authored or reviewed drafts of the article, and approved the final draft.

Okan Bilge Ozdemir analyzed the data, authored or reviewed drafts of the article, and approved the final draft.

Kevser Özdem Karaca conceived and designed the experiments, analyzed the data, authored or reviewed drafts of the article, and approved the final draft.

Nuh Azgınoğlu analyzed the data, authored or reviewed drafts of the article, and approved the final draft.

Ethics

The following information was supplied relating to ethical approvals (i.e., approving body and any reference numbers):

Gazi University Clinical Research Ethics Board granted Ethical approval to carry out the study within its facilities (Decision No. 616).

Data Availability

The following information was supplied regarding data availability:

The Gazi Brains 2020 dataset is available at Kaggle:

https://www.kaggle.com/datasets/gazibrains2020/gazi-brains-2020.

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
