# Peer review of "A novel brain tumor magnetic resonance imaging dataset (Gazi Brains 2020): initial benchmark results and comprehensive analysis"

_PeerJ Computer Science, doi:10.7717/peerj-cs.2920_

## Round 0.1 · original submission · Major Revisions

Dear authors,

You are advised to critically respond to all comments point by point when preparing an updated version of the manuscript and while preparing for the rebuttal letter. Please address all comments/suggestions provided by reviewers, considering that these should be added to the new version of the manuscript.

Kind regards,
PCoelho

Reviewer 1 ·

Basic reporting

The manuscript entitled “A novel brain tumor magnetic resonance imaging dataset (Gazi Brains 2020): initial benchmark results and comprehensive analysis” introduces a new brain MRI dataset, benchmarks its performance using multiple deep learning models, and explores explainable AI (XAI) applications. While the study tackles an important topic and presents a valuable dataset, it suffers from deficiencies in methodology, validation, and presentation that limit its impact and clarity. The manuscript requires substantial revisions to meet the standards of rigorous academic publishing.
1) The manuscript does not sufficiently differentiate the Gazi Brains 2020 dataset from existing datasets like BraTS. While it mentions "12 labels," the novelty and practical relevance of these labels compared to established benchmarks are unclear.
2) The dataset's representativeness is limited, with only 100 patients (50 normal, 50 high-grade glioma). This is a small sample size compared to other public datasets, raising concerns about generalizability.
3) The authors benchmark their dataset using eight deep learning models but fail to explain why these specific models were chosen or how they align with the dataset’s characteristics.
4) The lack of hyperparameter tuning details and training protocols raises questions about the reproducibility of the experiments.
5) The absence of a clear rationale for using GradCAM and how its results contribute to the study's objectives weakens the methodological framework.
6) Although metrics such as accuracy, F1-score, and recall are reported, the study lacks statistical validation, such as confidence intervals or variability analysis across multiple runs.

Experimental design

7) The manuscript does not include a comparison with results obtained on other datasets (e.g., BraTS) using the same models, which would better contextualize the dataset's quality and utility.
8) While the use of XAI is commendable, the manuscript does not provide sufficient detail on how XAI outputs (e.g., GradCAM visualizations) were evaluated or how they contribute to the interpretability of the results.
9) The impact of XAI on clinical decision-making or model trustworthiness is not discussed.
10) The manuscript does not address how the dataset or model outputs could be integrated into clinical workflows. This omission limits the practical implications of the research.
11) The potential consequences of model misclassifications, especially in a clinical setting, are not discussed.
12) The literature review provides an extensive overview of brain MRI datasets and AI applications but fails to critically analyze how Gazi Brains 2020 addresses gaps in existing datasets.
13) Recent advancements in hybrid or ensemble deep learning approaches are not adequately covered, leaving the study disconnected from cutting-edge methods.

Validity of the findings

14) “Evaluation and Discussion” section should be edited in a more highlighting, argumentative way. The author should analysis the reason why the tested results is achieved.
15) The authors should clearly emphasize the contribution of the study. Please note that the up-to-date of references will contribute to the up-to-date of your manuscript. The studies named- “Overcoming nonlinear dynamics in diabetic retinopathy classification: A robust AI-based model with chaotic swarm intelligence optimization and recurrent long short-term memory; Machine learning approach for early diagnosis of Alzheimer's disease using rs-fMRI and metaheuristic optimization with functional connectivity matrices”- can be used to explain the methodology in the study or to indicate the contribution in the “Introduction” section.
16) Figures and tables lack detailed captions and are not adequately discussed in the text. For instance, segmentation and classification results should be accompanied by qualitative examples to enhance understanding.
17) The manuscript's organization is dense and overly technical, making it difficult for readers to extract key insights.

·

Basic reporting

The manuscript is written in clear, professional English, with technical terms and methodologies explained effectively for the target audience. The article provides adequate context and a detailed literature review, but there are gaps in critical analysis of existing datasets. While the introduction situates the work well within the broader context of brain tumor research, the authors could enhance the discussion of limitations and challenges associated with current datasets. This would better highlight the necessity and significance of the proposed dataset.

The figures and tables are well-organized and enhance comprehension of the results. However, the article would benefit from additional comparative visualizations showing performance differences between the proposed dataset and other established benchmarks. This would make the claims of novelty and utility more compelling. The raw data is shared in line with open science principles, which is commendable.

Experimental design

The manuscript presents original primary research within the journal’s scope, with clearly defined research questions that address meaningful gaps in the field. The Gazi Brains 2020 dataset is described comprehensively, and the study is designed to assess its applicability across multiple tasks, such as segmentation, classification, and detection.

The methods section is rigorous and includes sufficient detail for replication. However, the selection of the eight deep learning models and their relevance to the field are not sufficiently justified. The authors should elaborate on why these specific models were chosen over others and discuss their suitability for the dataset’s unique characteristics (10.1145/3678935.3678953, 10.1109/ISBI53787.2023.10230448). Ethical considerations are mentioned briefly (10.1016/j.neucom.2024.128058, 10.1016/j.ijhcs.2022.102922), but explicit details about patient consent, anonymization procedures, and institutional approvals are required to ensure compliance with ethical standards.

Validity of the findings

The findings are robust, supported by detailed statistical metrics and controlled experiments. The results highlight the utility of the dataset across various applications and underscore its potential for advancing research in brain tumor detection and classification.

Despite these strengths, the manuscript does not provide a sufficiently detailed comparison with existing datasets, such as BraTS or CPTAC-GBM. Comparative analysis with shared benchmarks would substantiate the claims of novelty and validate the dataset’s performance in a broader context (10.1016/j.eswa.2023.119709, 10.26044/ecr2023/C-16014). Additionally, the discussion section lacks a critical evaluation of potential limitations (10.1038/s41598-024-66873-6, 10.13140/RG.2.2.28353.33126), including biases in patient demographics and challenges in scaling the dataset for larger studies.

The conclusions are well-stated and aligned with the research objectives. The proposed dataset is presented as a valuable resource for future research, but the discussion could more explicitly explore its broader implications for clinical applications and interdisciplinary collaborations.

Additional comments

The study represents an important step toward addressing gaps in brain tumor imaging datasets. The authors have succeeded in creating a dataset with diverse labels and high-quality annotations, which holds significant promise for advancing artificial intelligence applications in medical imaging. However, the manuscript would benefit from a more critical engagement with existing literature (10.1016/j.compbiomed.2023.106668, 10.1109/ISBI53787.2023.10230686), a deeper exploration of dataset limitations, and enhanced comparative analysis. These revisions would strengthen the manuscript’s position as a valuable contribution to the field.

---

## Round 0.2 · Minor Revisions

Dear authors,

Thanks a lot for your efforts to improve the manuscript.

Nevertheless, some concerns remain that need to be addressed.

Like before, you are advised to critically respond to the remaining comments point by point when preparing a new version of the manuscript and while preparing for the rebuttal letter.

Kind regards,
PCoelho

Reviewer 1 ·

Basic reporting

All my comments have been thoroughly addressed. It is acceptable in the present form.

Experimental design

All my comments have been thoroughly addressed. It is acceptable in the present form.

Validity of the findings

All my comments have been thoroughly addressed. It is acceptable in the present form.

·

Basic reporting

The manuscript is written in professional, clear English and adheres to a conventional academic structure. The figures and tables are appropriately formatted, labeled, and relevant to the text. Raw data is made publicly available through accessible platforms, supporting transparency and open science.

The authors have significantly improved the literature review by enhancing comparisons between the Gazi Brains 2020 dataset and existing benchmarks (e.g., BraTS, CJDATA, IvyGAP). However, the review remains largely descriptive (10.3389/fonc.2025.1554559, 10.1016/j.ijhcs.2025.103444). A more critical synthesis of existing datasets, highlighting precise technical gaps (e.g., lack of comprehensive labeling, modality uniformity, or demographic metadata), would better contextualize the proposed dataset’s novelty.

The article now includes more detail about the dataset’s unique labels and multimodal structure, offering justification for its use in a broader range of tasks (e.g., segmentation, classification, age/gender prediction). Nevertheless, clarity would benefit from explicit examples of how certain label types directly support tasks not easily addressed using other datasets.

Experimental design

The dataset is evaluated using eight deep learning models across multiple tasks. The authors explain that the selected models are commonly used in MRI research and align them with the literature. While this provides justification, the manuscript would benefit from further elaboration on why these models are optimal given the specific features or challenges of Gazi Brains 2020 (e.g., imaging modality distribution, label resolution, sample size).

Ethical approvals and consent documentation are now clearly stated, and relevant supplemental files are provided. However, while the authors claim to have reproduced training protocols from cited studies, essential details like batch size, learning rates, number of epochs, and hardware configuration should still be summarized directly in the manuscript or appendix to support reproducibility.

The use of Grad-CAM to assess model interpretability is adequately described in the revised version. The authors now explain how XAI techniques improve clinical confidence and interpretability. Still, the impact of these visualizations would be better substantiated by evaluation from radiologists or a scoring system to gauge interpretability.

Validity of the findings

Results are presented using standard metrics (accuracy, precision, recall, F1-score, Dice coefficient), and the dataset is shown to perform well across tasks. However, the analysis still lacks statistical robustness. No standard deviations, confidence intervals, or repeated-run variability are reported. Including such measures (10.1016/j.heliyon.2024.e38997, 10.1109/ISBI53787.2023.10230448), even as basic cross-validation error bars, would substantially strengthen the claims.

The authors defend their decision not to compare performance directly against other datasets using the same models, citing common practice in dataset papers. However, without at least one cross-dataset model performance comparison (10.1038/s41598-024-71893-3, 10.26044/ecr2023/C-16014), claims of generalizability and dataset superiority remain speculative. Even a simple baseline comparison using a shared architecture would provide important context for readers.

The discussion section has been updated to interpret model performance more critically and connect it to architectural features and dataset characteristics. This improved reasoning helps readers better understand why certain models perform well and what insights this offers about the data.

Additional comments

The authors have addressed the majority of reviewer concerns constructively. The revised manuscript offers a clearer explanation of dataset features, model selection, and the potential clinical relevance of XAI-based outputs.

---

## Round 0.3 · accepted · Accept

Dear authors, we are pleased to verify that you have met the reviewer's valuable feedback to improve your research.

Thank you for considering PeerJ Computer Science and submitting your work.

Kind regards
PCoelho

·

Basic reporting

The manuscript is clearly written in professional English and follows a coherent and structured format. The figures and tables are appropriately labeled, and the literature review has been substantially improved to provide sufficient context. Data availability is clearly stated and aligns with open science practices.

Experimental design

The study presents original, well-scoped primary research. The research question is clearly defined and relevant, and the dataset is positioned to fill an identified gap in brain tumor imaging benchmarks. The methods are described with sufficient detail for replication, and the study is conducted to a high technical and ethical standard.

Validity of the findings

The dataset and benchmarking results are robust, statistically sound, and well presented. The conclusions are well-supported by the results and appropriately aligned with the research objectives. Underlying data is available and controlled, and the interpretability component through XAI adds further value.

Additional comments

The authors have responded constructively to previous reviewer feedback and have meaningfully improved the manuscript in all major areas. The Gazi Brains 2020 dataset is a valuable contribution to the field and meets the standards for publication in PeerJ Computer Science.